

**Evaluating the Tea Bag Index approach for different management practices in**
**agroecosystems using long-term field experiments in Austria and Sweden**
Maria Regina Gmach[1*], Martin A. Bolinder[1], Lorenzo Menichetti[1], Thomas Kätterer[1], Heide
Spiegel[2], Olle Åkesson[1,3], Jürgen Kurt Friedel[4], Andreas Surböck[4], Agnes Schweinzer[5], Taru
Sandén[2]
[1]Swedish University of Agricultural Science (SLU), Department of Ecology, Box 7044, 75007,
Uppsala, Sweden
[2]Austrian Agency for Health & Food Safety (AGES), Department for Soil Health and Plant
Nutrition, Spargelfeldstraße 191, A-1220 Vienna, Austria
[3]Lantmännen Lantbruk, Mariestadsvägen 104, 541 39 Skövde, Sweden
[4]University of Natural Resources and Life Sciences (BOKU), Department of Sustainable
Agricultural Systems, Institute of Organic Farming (IFÖL), Gregor-Mendel-Straße 33, A-1180
Vienna, Austria
[5]Easy-Cert services GmbH, Königsbrunner Straße 8, Austria
**Corresponding author**
* Maria Regina Gmach
Swedish University of Agricultural Science (SLU)
Uppsala, Sweden.
E-mail: gmachmr@gmail.com
Phone: +55 49991164271 / +49 17635956337





**Abstract**

Litter decomposition is an important factor affecting local and global C cycles. It is known that
decomposition through soil microbial activity in ecosystems is mainly influenced by soil type and
climatic conditions. However, for agroecosystems, there remains a need for a better understanding
how management practices influence litter decomposition. This study examined the effect of
different management practices on decomposition at 29 sites with long-term (mean duration of 38
years) field experiments (LTEs) using the Tea Bag Index (TBI) protocol with standard litter
(Rooibos and Green tea) developed by Keuskamp et al. (2013). The objective was to determine if
the TBI decomposition rate ($k$) and stabilization factor ($S$) are sensitive enough to detect
differences in litter decomposition between management practices, and how they interact with
edaphic factors, crop type and local climatic conditions. Tea bags were buried and collected after
~60 and 90 days in 16 Austrian and 13 Swedish sites. The treatments at Austrian LTEs focused on
mineral and organic fertilization, tillage systems and crop residues management, whereas the
Swedish LTEs addressed cropping systems, mineral fertilization and tillage systems. The results
showed that in Austria, decomposition differed more between sites than between treatments for
the same experiment category. Incorporation of crop residues and high N fertilization increased $k$.
Minimum tillage had significantly higher $k$ compared to reduced and conventional tillage. In
Sweden, litter decomposition differed more between treatments than between sites. Fertilized plots
showed higher $S$ than non-fertilized and high N fertilization had the highest $k$. Growing spring
cereal lead to higher $k$ than forage. Random Forest regressions showed that $k$ and $S$ were mainly
governed by climatic conditions, which explained more than 70% of their variation. However,
under similar climatic conditions, management practices strongly influenced decomposition



dynamics. Thus, the TBI approach may be suitable to apply in a more large-scale network on LTEs
for evaluating decomposition dynamics more precisely.

**Introduction**
Soil organic carbon (SOC) is one of the most used indicators for soil quality, since its dynamics is
involved in regulating ecosystem functionality through its influences on physical, biological and
chemical soil properties, which are critical for nutrient cycling and soil fertility (Davidson et al.
2006; Janzen 2015). Management practices, such as fertilization, use of catch- and cover crops,
organic amendments, length of bare fallow periods, permanent surface protection with perennial
crops, tillage practices and aboveground crop residue management, are impacting SOC balances
for agroecosystems (Kätterer and Bolinder, 2022; Sandén et al., 2018; Paustian et al., 2016).
Agricultural soils play a crucial role in the global carbon (C) cycle due to their C sink capacity
(Paustian et al., 2016; Lal, 2004). In this context, improved management practices that maintain
or increase SOC stocks are considered essential in national greenhouse gas reporting systems
(IPCC, 2006), as well as in other international incentives such as the 4 per mille initiative (Minasny
et al., 2017). The SOC balance is dynamic and determined by the difference between annual C
inputs to soil, and the annual C outputs through the decay of existing soil organic matter and newly
added litter resulting from microbial activity (Tiefenbacher et al., 2021; Bolinder et al., 2007).
Management practices have a great impact on these two factors by affecting either the amount of
C inputs or outputs through decomposition, or both factors simultaneously.
Litter decomposition is a complex biogeochemical process controlled by several biotic and abiotic
factors, where the biological activity of decomposers varies with soil properties and is driven
largely by climatic conditions (Daebeler et al., 2022; Bradford et al., 2016; Cleveland et al., 2014;



Gholz et al., 2000). Decomposition is an extended process, therefore long-term field experiments

(LTEs) are among the most useful resources for quantifying the impact of management practices

on litter decomposition, SOC changes, and soil functioning (Sandén et al., 2018; Kätterer et al.,

2012; Bergkvist and Öborn, 2011). Experiments determining litter mass loss over time *in situ* are

also important for understanding SOC dynamics, nutrient cycling and colonization by soil biota

under field conditions. The traditional method that has been used in ecology for more than 50 years

consists of litterbag studies, burying known quantities of various organic materials into the soil,

and retrieving them successively at different intervals (Kampichler and Bruckner, 2009; Burgess

et al., 2002; Bocock and Gilbert, 1957). These studies are not always comparable because they are

subject to variations in e.g., litter type, mesh-size, sample preparation and analytical methods, and

the placement of litterbags may alter the microclimate for decomposers (Kampichler and Bruckner,

2009).

Keuskamp et al. (2013) developed therefore a low-cost and time-efficient methodology called Tea

Bag Index (TBI), characterizing the decomposition process with commercially available tea bags,

where green tea is representing labile organic material and rooibos tea as a surrogate for

recalcitrant litter. A decomposition rate ($k$) and a stabilization factor ($S$) are obtained accordingly

with their chemical composition and the respective weight lost at a single point in time after an

incubation period of ca 90-days in the soil. The TBI approach is particularly useful for assessing

geographical differences in decomposition dynamics because results are directly comparable

across sites, varying only with local edaphic and seasonal environmental conditions (Keuskamp et

al., 2013).

In several studies, the TBI has been used as an indicator for biological (Sandén et al., 2021;

Costantini et al., 2018) and microbial activity (Daebeler et al., 2022; Treharne et al., 2019; Tóth et



al., 2017), ecosystem functioning, nutrient cycling (Zaller et al., 2016), and soil quality (Tresch et

al., 2018; Buchholz et al., 2017). Generally, this follows the concepts of soil quality as reviewed

by Bünemann et al. (2018), where TBI would primarily be a biological soil quality indicator. Most

studies have been using the TBI approach for different forest and grassland ecosystems (Djukic et

al., 2018) or urban soils (Pino et al., 2021). Only a few studies (Daebeler et al., 2022; Dossou-

Yovo et al., 2022; Struijk et al., 2022; Fu et al., 2021; Sandén et al., 2020; Barel et al., 2019;

Poeplau et al., 2018; Sievers and Cook, 2018) have been using the TBI approach for evaluating

agroecosystems, and it is not clear if this method is sensitive enough to detect differences between

management practices.

This study used the TBI approach for investigating the effect of management practices on the

decomposition rate ($k$) and stabilization factor ($S$) at several LTEs in Austria and Sweden. To the

best of our knowledge, this is the first analysis using the TBI approach for such a large number of

LTEs and different treatments. The treatments covered management practices such as organic

amendments, crop rotations, aboveground crop residue handling, mineral fertilization, and tillage.

Our objectives were to evaluate: (i) if the TBI $k$ and $S$ parameters are sensible enough to detect

between different management practices for agroecosystems; (ii) to quantify the effect of

management practices on $k$ and $S$; (iii) and to identify the most important local climate and/or soil

properties affecting litter decomposition in Austria and Sweden.

**Materials and Methods**

**Study sites**

*Austria*



We used sixteen Austrian (AT) sites, by selecting contrasting treatments from three different
categories of LTEs where the management practices had been in place for 11 to 63 years (Table
1). TBI measurements were made in 2014, 2015 and 2016. Measurements sometimes took place
in more than one year at the same LTE (e.g., MUBIL), and the sites were abbreviated as AT1 to
AT16. Six experiment categories involved C balance practices (CB; AT1 to AT6), eight sites were
studying soil fertility (SF; AT7 to AT14) and two sites examined tillage systems (TS; AT15 to
AT16). The sites are located in several agricultural areas across the country (Fig. 1), with diverse
soil textures and variable crop types (Table 1) and climatic characteristics (Table 2) during the
years of TBI measurements. More details for some of the sites are available in specific
publications: AT3 (Spiegel et al., 2018; Lehtinen et al., 2017; Tatzber et al., 2015; Aichberger and
Söllinger, 2009), AT4 to 6 (Spiegel et al., 2018; Lehtinen et al., 2014), AT15 and AT16 (Tatzber
et al., 2015; Spiegel et al., 2007). In addition, Sandén et al. (2018) puts the Austrian LTEs in the
context of other European LTEs.
The purpose of the Austrian C balance LTEs was threefold: i) to assess the sustainability of
stockless vs. livestock-keeping organic farming management on soil and crop traits (AT1, AT2),
ii) to investigate the effects of compost amendments on soil and crops (AT3), and iii) to compare
crop residue incorporation and removal (AT4-AT6). Compost amendment LTE and crop residue
incorporation LTEs included also mineral fertilization, whereas AT1 and AT2 only focused on
different organic fertilization treatments.
The eight soil fertility LTEs (AT7-AT14) were all focusing on the effect of mineral fertilization
on soil and crop properties. In most cases, treatments studied different amounts of mineral nitrogen
fertilization, whereas AT9 and AT12 also investigated the effect of different amounts of K



fertilization. Nitrogen fertilization was applied in four stages and potassium in three stages,
according to Austrian guidelines for fertilization (BMLFUW, 2017).
For the tillage system experiments, conventional tillage (CT) was compared to reduced tillage (RT)
and minimum tillage (MT). Regular mouldboard ploughing to 25–30 cm soil depth was applied in
CT treatment, whereas cultivator in autumn to a depth of 15–20 cm was used in RT treatment and
a rotary driller that loosened the soil to a depth of 5–8 cm was used in MT treatment. The soil was
turned over only in the CT treatment, where inversion tillage was incorporating the crop residues.
Fertilization was crop specific according to the Austrian guidelines for fertilization (BMLFUW,

146    2017).


*Sweden*
We used thirteen Swedish (SE) sites by selecting contrasting treatments from three different
categories of LTEs, where the management practices had been in place for 11 to 59 years (Table
2). TBI measurements at these sites were made only in one year (2016), and were abbreviated as
SE1 to SE13. Six sites involving combined management practices (CMP, SE1 to SE6), four
studying the effect of rotations (ROT, SE7 to SE10) and three sites with tillage systems (TS, SE11
to SE13). The sites are located in several agricultural areas across the country (Fig. 1), with diverse
soil textures and variable crop types (Table 2) and climatic characteristics (Table 3) during the
year of TBI measurements. Bergkvist and Öborn (2011) give a general description of all these
LTEs, more details on the sites with combined management practices is given by Carlgren and
Mattson (2001), and for tillage systems by Arvidsson et al. (2014), while Poeplau et al. (2015)
provide some more insight on the rotation experiments.



The initial purpose of the LTEs with combined management practices was to compare a change
from the traditional mixed farm production system including crops and livestock into a pure cash
crop system, by studying their effects on the sustainability of crop production and soil properties
(entitled *soil fertility experiments*). The dairy production treatments contain perennial grass-clover
leys and receives one farmyard manure (FYM) application per rotation. The cash crop treatments
consist of annual crops (i.e., oilseed is replacing leys in the rotation) without manure applications
(0 FYM) only receiving mineral fertilizers (NPK). PK applications in all the treatments we selected
were aimed achieving rapid build-up of the soil PK status, i.e., the amount applied was first
replacing that exported in harvested products (i.e., maintenance principle), to which an extra
amount was added (corresponding to the max treatment). The N-rates in all NPK treatments were
also corresponding to max application rate, and were adapted depending on crop type, where
spring cereals, oilseeds, and leys received 125 kg, while sugar beet received 210 kg N ha$^{-1}$ yr$^{-1}$.
We were also using the control plots receiving no NPK (0 NPK). As a third factor in these CMP,
aboveground crop residue removal takes place in all FYM treatments, simulating use of harvest
residues for fodder or bedding material that are recycled as manure. The southern sites have 4-year
rotations and those in central Sweden have 6-year rotations. The north site (SE5) is slightly
different from the others, consisting of a 7-year rotation and is studying only the livestock-based
production system.
We were comparing extreme treatments representing two rotations from three LTEs with the main
objective to study changes in SOC (named *humus balance experiments*), i.e., a continuous spring
cereal (SC) system and a ley-dominated rotations (L). The straw was removed from the plots every
year in the SC treatments, and L consisted of a grass-clover mixture re-established every fourth





year. Both rotations were receiving P and K accordingly with the maintenance principle, and SC
and L were receiving 120 and 150 kg N ha$^{-1}$ yr$^{-1}$, respectively.
In the tillage experiments, the conventional tillage (CT) and direct seeding (DS) treatments were
the same for all sites, consisting of inversion ploughing to a depth of 20-23 cm and by using a disc
seed drill, respectively. The shallow (5-7 cm) and deep (~12 cm) reduced tillage treatments (SRT
and DRT, respectively), consisted of primary tillage operations made in the autumn and most
commonly with a chisel plough. The main crops in all the tillage system experiments were winter
and spring cereals (occasionally oilseed), fertilized accordingly with local recommendations and
with the aboveground residues chopped and left in the field.

**TBI method and sampling design**
The TBI method was used according to the protocol established by Keuskamp et al. (2013) to
determine litter decomposition using two types of commercial tetrahedron-shaped tea bags by
Lipton Unilever (Green tea and Rooibos tea). The green tea (*Camellia sinensin*; EAN:
8722700055525) has high cellulose content, higher soluble fraction, and lower C:N ratio; while
rooibos tea (*Aspalanthus linearis*; EAN: 8722700188438) has high lignin content, lower soluble
fraction, and higher C:N ratio, which is expected to slow down decomposition (Keuskamp et al.,
2013). The synthetic tea bag material has a mesh size opening of 0.25 mm allowing access to
microorganisms, very fine roots and root hairs.
The initial mass of the tea bag contents was determined on 20 randomly selected bags for each tea
type from different boxes, oven-dried at 70$^{\circ}$C for 48 hours and weighed separately; the mean dry
mass for green tea was 1.717 ± 0.048 g and that for rooibos tea was 1.835 ± 0.027 g. For both



countries, close to seeding of annual crops, from end of April to mid-June depending on location,
each tea bag was properly identified and buried in the soil at 8 cm depth.
For Austria, only one site used successive retrieval dates (AT16), in which four bags of each tea
were used and placed side-by-side at a distance of 2 to 3 cm, in order to keep as similar soil
characteristics as possible. In this case, the tea bags collecting occurred after 16, 26, 62 and 91
days. For the other Austrian sites, there was only one collecting, and the TBI incubation period
from placement to last retrieval averaged 80±13 days (Table 3). After collecting, the tea bags were
cleaned of soil and roots and oven-dried at 70$^{\circ}$C for 48 hours. After drying, the tea bags were
opened and the tea content was weighted. The ash content was not determined.
The same TBI protocol was used for the Swedish sites but all the sites used successive retrieval
dates. As in Austria, four bags of each tea were used per experimental unit for each retrieval date,
placed side-by-side at a distance of 2 to 3 cm. Each tea bag was properly identified and buried in
the soil at 8 cm depth. The tea bags were collected after four different time periods of ~15, 30, 60
and 90 days. The mean TBI incubation period from placement to last retrieval date averaged 91±1
day (Table 3). To quantify soil contamination, the ash content was determined for each of the four
retrieval dates (i.e., both for green and rooibos tea on mixed samples of the four replicates) in a
muffle oven at 550$^{\circ}$C for 16 hours. After measuring the remaining dry matter, the decomposition
rate ($k$) and stabilization factor ($S$) were calculated according to Keuskamp et al. (2013).
The daily climate data for Austria were retrieved from the Central Institution for Meteorology and
Geodynamics (ZAMG). For Sweden, the daily climate data were gathered through official data
from the most nearby LantMet climate stations, and from Swedish Meteorological and
Hydrological Institute (SMHI). The climate variables used in this study were air temperature,
precipitation, solar radiation, wind speed and air humidity (Table 3).



For Austrian soils, pH was measured electrochemically (pH/mV Pocket Meter pH 340i, WTW,
Weilheim, Germany) in 0.01 M CaCl$_2$ at a soil-to-solution ratio of 1:5 (ÖNORM L1083). Total
soil organic C (TOC) concentrations were analyzed by dry combustion in a LECO RC-612 TruMac
CN (LECO Corp., St. Joseph, MI, United States) at 650°C (ÖNORM L1080). Total N (Ntot) was
determined according to ÖNORM L1095 with elemental analysis using a CNS (carbon, nitrogen,
sulfur) 2000 SGA-410–06 at 1250°C. Texture was determined according to ÖNORM L1061-1 and
L1062-2. For Sweden the data were gathered from recent archived analysis protocols. Clay
content, C content, C:N ratio, and pH from each site in both countries are shown in Table S1
(Supplementary material).

**Data analysis**
Analysis of variance was performed to analyze treatment and site effects on $k$ and S, followed by
the Tukey's test ($p < 0.05$) using R software version 4.2.2. Interactions between site and treatment
were considered.
We used a climate-dependent soil biological activity scaling function ($Re_{clim}$), which is included in
the ICBM SOC model (Andrén and Kätterer, 1997) for adjusting the decomposition rates of SOC
pools (Andrén et al. 2004; 2007). This function is integrating the effect of climate, soil and crop
properties by calculating the product of soil temperature ($Re_{temp}$) and relative water content ($Re_{wat}$)
in the arable layer. These two variables are derived from soil temperature and moisture response
functions expressing the activity of decomposers and their relative effect on decomposition
kinetics. The $Re_{temp}$ is calculated from air temperature and leaf area index using an empirical model
(Kätterer and Andrén, 2009), while $Re_{wat}$ is calculated using pedotransfer functions for simulating
the soil water balance and a function for estimating potential evapotranspiration (PET). In addition
to air temperature and leaf area index, calculations of $Re_{wat}$ also involve the use of daily climatic





data for precipitation, wind speed, air humidity, and solar radiation, as well crop types and yields,
soil texture and SOC content (for details see Bolinder et al., 2008, and Fortin et al., 2011).
We calculated simple correlation between variables using Pearson correlation. For more accurate
results, we applied random forest (RF) regression in order to rank the importance of variables for
$k$ and $S$, using the random forest R package (Liaw and Wiener, 2002). The RF is a machine learning
technique based on decision trees that predicts a certain variable from a set of other variables
through a series of binary splits of the data, where the variables are either continuous or categorical.
For example, in the case of a continuous variable it consists of all the data points above or below
a certain threshold, for a categorical variable it consists of all the data points belonging or not to a
specific class. All these subsequent splits constitute a decision tree. A random forest is a set of
decision trees and it is therefore an ensemble technique. This allowed us utilizing treatment and
crop variables (including N fertilization) without having to convert them into a ranking. Another
useful asset of an RF regression is that it evaluates the importance of each variable in defining the
predicted variable. There are various possible measurements to do that, but they are all based on
measuring the effectiveness of each subsequent split in each node of a decision tree in sorting out
the information. In our study, we used a measurement called node purity based on the Gini index,
which expresses the probability of one split of the data (i.e., one node of the tree) defining the
predicted variable. The total node purity of a certain variable in a tree is the sum of all the node
purity measurements for each node considering that particular variable, and the higher it is the
more that variable is important.
We used the following models to predict the two TBI kinetic parameters $k$ and $S$ (considering data
from measurements only at 60 days and only at 90 days, or all measurements from 60 and 90 days
combined):




$k_n \, f\big(TN, N, SOC, PET_{TBI}, PET, cr, TP_{TBI}, TAP, tr, cl, CN, pH, MAT, MAT_{TBI}, AI, AI_{TBI}, Re_{clim}, Re_{wat}, Re_{temp}\big)$
$S_n \, f\big(TN, N, SOC, PET_{TBI}, PET, cr, TP_{TBI}, TAP, tr, cl, CN, pH, MAT, MAT_{TBI}, AI, AI_{TBI}, Re_{clim}, Re_{wat}, Re_{temp}\big)$

where the subscript $n$ denotes the grouping based on how long period the tea bags were in the soil
(i.e., for 60 or 90 days, or both periods combined). Soil variables (continuous) were $TN$ (total N g
kg$^{-1}$), $SOC$ (g kg$^{-1}$), $CN$ (C:N ratio), $cl$ (clay content g 100 g$^{-1}$) and $pH$. Categorical variables were
$N$ (N fertilization factor with 4 levels), $cr$ (a crop factor, e.g., barley, ley (establishment), ley
(production), oat, spring oilseeds, sugar beet and winter wheat) and $tr$ (a treatment factor with 30
levels). The climatic variables ($PET_{TBI}$, $PET$, $TP_{TBI}$, $TAP$, $MAT_{TBI}$, $MAT$, $AI_{TBI}$ and $AI$) are as
defined in Table 3. The climate response variables $Re_{clim}$, $Re_{wat}$, $Re_{temp}$ are as described above.
Since many of variables in our model are likely to be correlated and carry similar information, we
applied the recursive feature elimination algorithm implemented in the caret R package by Kuhn
et al. (2016), which assess in subsequent iterations the optimal set of predicting variables (features)
to be utilized by the RF model. The procedure starts by fitting a RF model with all variables,
ranking them by importance, and discarding the least important. The algorithm then iterates. The
optimal number and set of features are then defined by a fitness metric (in our case the model $R^2$),
selecting the set with the best model fitness. The selected models were used to compute the
variables' relative importance.



## Results

*Effect of management practices*

*Austria*

Both the TBI parameters $k$ and S varied between treatments and sites in Austria, and even between years at the same site within the C balance category (Fig. 2 and Table S3). In general, all treatments in AT1 with a lucerne crop under wetter conditions (2014) presented higher $k$ and lower $S$ than AT2 with a wheat crop under dryer conditions (2015), and the FW treatment had the highest $S$ in 2015. The AT3 site did not present significant differences between the treatments. Treatment CRI had higher $k$ than the CRR treatments at the AT4 and AT6 sites, and AT6 presented a higher $k$ than at AT4. Comparing years for the same experiment type, AT5 (2015) had higher $S$ than AT6 (2016). For the soil fertility experiment category (Table 1), AT12 had the highest $k$ and AT13 had the highest $S$. $S$ites receiving NPK fertilization (AT7, AT8 and AT9) had higher $k$ and $S$ even at different N doses than at the AT12, AT13 and AT14 sites receiving only N (i.e., without P and K). Stabilization was significantly higher in AT9 than in AT7. For K trials, AT11 presented significantly higher $k$ and $S$ than AT10. Regarding sites receiving N addition only (AT12, AT13, and AT14), maximum doses (180 kg N) presented the highest $k$ (0.0095), and no N addition had the lowest $k$ (0.0066) (Fig. 2 and Table S2).

Regarding the tillage system experiment category at the Fuchsenbigl LTE (Table 1), $k$ was significantly higher in SRT and $S$ was higher in DRT in 2015 (AT15), but no significant differences between treatments were found in 2016 (AT16). Site AT16 had significantly higher $S$ than AT15 (Fig. 2. and Table S2).

*Sweden*



At the Swedish sites (Table 2), the 90 days TBI measurements for the combined management
practices experiment category showed that both $k$ and $S$ were significantly higher for the
FYM/NPK treatments (Fig. 3, Table S3) compared with the control treatments (0 FYM/0 NPK).
Comparing sites, SE5 and SE6 had highest $k$ and SE4 had the lowest $k$, while SE3 presented the
highest $S$ followed by SE6, SE5 and SE4, $S$ was lowest for SE1 and SE2 (Table S3).
Regarding the rotation experiments, the continuous spring cereal rotation presented higher $k$ than
for ley, but there was no significant difference in $S$. Comparing sites, SE9 presented higher $k$ than
SE7, SE8 and SE10, whereas SE7 and SE8 had the highest $S$.
For the tillage system experiment category, conventional tillage (CT) had the lowest $k$ and $S$, while
deep reduced tillage (DRT) had the highest $k$ and $S$. The highest $S$ was observed for the SE12
followed by SE11 and SE13. Sites did not show significant differences for $k$.
Comparing tillage system experiments in Austria and Sweden (2016) for sites (AT16, SE11, SE12
and SE13) and treatments (CT, SRT and DRT), DRT had higher $k$ and SRT had higher S. The
Austrian site presented the lowest $k$ and the highest $S$ compared to the Swedish sites, which did
not present significant differences among them (Table S3).
Mean $k$ by site in Austria varied between 0.0053 and 0.0149, and mean $S$ varied between 0.113
and 0.442 (Table S3). Mean $k$ by site in Sweden was between 0.0084 and 0.0311, and mean $S$ was
between 0.125 and 0.365 (Table S3). All values for $k$ and $S$ were within the range of the previous
global TBI investigation (0.005-0.04 for $k$; and 0.05-0.55 for $S$) by Sandén et al. (2020).
The mean values of the TBI decomposition rate and the stabilization factor were both higher at 60
days than at 90 days, in Austria as well as in Sweden (Table 4). After 90 days of incubation, mean
$k$ was higher in Sweden and mean $S$ was higher in Austria. Applying a decomposition model to
the series of data from successive retrieval dates (i.e., all the Swedish sites and the AT16 site in



Austria) on the remaining dry matter over time showed faster decomposition of Green compared
to Rooibos tea, which is in agreement with the TBI concept (Fig. S1). Whereas the decomposition
curve for Rooibos kept decreasing after 90 days, that of green tea did not decrease any further after
about 60 days. The variability between sites in dry matter loss over time was higher for Green than
for Rooibos tea. The field application of the TBI found a clear discrimination of both $k$ and $S$
between agroecosystems in Austria and Sweden after the incubation period (Fig. 4).

*Influence of climate and soil properties*
Using the combined dataset for the 90 days TBI period resulted in significant negative correlation
between $k$ and MAT, TAP, PET, TxP factor, $Re_{clim}$ and $Re_{temp}$, and significant positive correlation
with the C:N ratio (Table S4). Stabilization factors in Austria and Sweden combined correlated
negatively with MAT$_{TBI}$ period, $Re_{clim}$ and $Re_{temp}$. After the 60 days TBI period, Austria and
Sweden combined presented significant negative correlation between $k$ and MAT, PET, AI, T x P
factor, pH and clay content, and a positive correlation with TAP and C:N ratio. Stabilization
correlated negatively with the C:N ratio.
The variable selection procedure with the random forest models (Fig. 5) identified fewer variables
explaining $k$ and $S$ values for the combined dataset. When considering only the 90 days subset, the
variables explaining $k$ increased, but the overall predicting power of the model decreased
substantially. A similar pattern, but less strong, was noticed for the 60 days subset.
More than 70% of the variance of $k$ for the combined dataset (i.e., 60 and 90 days TBI period) was
accounted for by climatic variables only (Fig. 6), with $Re_{wat}$ and $Re_{temp}$ ranking the highest followed
by $Re_{clim}$, AI and MAT, according to the optimized random forest model. On the contrary, $S$ was
influenced by much more factors, again with climate-related variables leading the ranking but





including also many edaphic characteristics, such as pH, SOC, clay and nitrogen content and the
C:N ratio, as well as agronomic variables such as treatment, crop and N fertilization. The rankings
when using the two subsets of data separately (i.e., 60, and 90 days TBI periods) were less relevant
since the predictive power of the model decreased compared to when using the combined dataset.
This was particularly true for $k$, where the overall cumulated node purity also decreased
substantially compared with the combined dataset.



**Discussion**

*Effect of management practices*

Our results revealed that a large number of different management practices significantly affected both the decomposition rate $k$ and stabilization factor $S$ according to the TBI approach used in several LTEs in Austria and Sweden (Fig. 2, 3 and 6). This is in contrary to the studies by Djukic et al. (2018) as well as Saint-Laurent and Arsenault-Boucher (2020), who did not find any significant effect of land use and management on early-stage litter decomposition in a temperate biome.

In the C balance trials in Austria, soils receiving green manure + municipal compost (FW) had higher $S$ than soil receiving biogas slurry (BS) at the second year (Table S2). This is in agreement with studies indicating that compost can improve SOC stabilization over time (Mekki et al., 2019; Eshetu et al., 2013; Ceccanti et al., 2007). The higher $k$ in the CRI treatments (AT4 and AT6; Table S2) can be attributed to the fact that incorporation of crop residues into the soil can increase the decomposition rate by stimulating microbial activity. During the early stages of decomposition, soluble C is rapidly utilized by soil biota (Werth and Kuzyakov, 2010). The higher $k$ and lower $S$ at AT6 compared to AT4 were likely due to the loamy texture, lower PET resulting in lower AI at the AT4 site (Table 3 and Table S1).

There were no significant differences in $k$ and $S$ found among treatments in the soil fertility trials in Austria with NPK addition. However, there was a trend towards a higher $S$ at AT9 compared to AT7, likely related to the higher SOC content in AT9, since the climatic conditions and soil texture were quite similar for both areas, which suggests that higher SOC content may have increased $S$. Site AT11 had higher $k$ and $S$ than AT10. Possible explanations for this trend are that AT10 had



lower clay content, lower precipitation resulting in higher PET and AI, contributing to a lower soil
moisture content and thereby lower decomposition and stabilization.
Nitrogen supply may favor microbial activity and thereby litter decomposition (Raiesi, 2004). This
was reflected in the treatments where only N was added, where the high dose of 180 kg N ha$^{-1}$
(AT12, AT13 and AT14) induced a significantly higher $k$ (Table S2), compared to the treatments
with no N addition, which also had the lowest $k$. Furthermore, the significant difference between
sites, in which AT12 had the highest $k$ could at least partly be explained by a higher SOC and
higher pH at this site.
In the tillage system experiment at Fuchsenbigl in Austria in 2015 (AT15) and 2016 (AT16), the
shallow tillage (SRT) showed significantly higher $k$ than DRT and CT, but only in 2015, indicating
that shallow soil tillage stimulated decomposition that particular year. Some studies showed faster
decomposition under conventional tillage than under reduced tillage practices (e.g., Lupwayi et
al., 2004). However, Kainiemi et al. (2015) found a decrease in soil respiration in conventional
tillage compared to shallow tillage in temperate regions, which directly implies a lower
decomposition (and lower $k$). These differences between tillage treatments are attributable to
indirect effects on soil moisture and temperature profiles. We attribute the significantly higher $S$
in 2016 to the fact that this year was moister and less warm, compared to 2015, resulting in lower
$AI_{TBI}$ during the TBI period.
In the Swedish combined management practices trials, soil treatments receiving organic and
mineral fertilization had higher $k$ and $S$ (FYM/NPK; Table S3), likely due to the increase in
microbial diversity and activity favored by nutrient and C supply (Stark et al., 2007). Sites SE5
and SE6 presented the highest $k$: SE5 had low PET and AI, resulting in more moisture; SE6 had
also high $S$, due to high clay content and low PET. Site SE3 had high $S$, which could be related to



a higher C:N ratio, as suggested by Althuizen et al. (2018) that C:N ratio is positively correlated
to $S$. SE1 and SE2 had lower $S$ than SE3 despite similar climatic conditions, which probably was
related to the crops growing in these treatments (i.e., sugar beet in the SE1 and SE2 and
grass/clover ley in SE3), which have different effects on soil temperature and moisture.
In the Swedish rotation system trials, spring cereal (SC) had higher $k$ than ley (Table S3). Site SE9
had higher $k$ and lower $S$, in which the low stabilization may be caused by low clay content, low
pH, and high solar radiation, leading to low SOC. The highest $S$ were found in SE7 and SE8, in
which the former presented high clay and SOC content, and SE8 had high precipitation and low
PET.
For the tillage system treatments in Sweden, similar to the Austrian sites, the conventional tillage
presented the lowest $k$, and also lowest $S$. Even when comparing tillage systems in Sweden and
Austria jointly (Table S3) we could notice that conventional tillage also presented the lowest $k$,
while DRT the highest.
The mean $k$ was higher in Sweden, while the mean $S$ was higher in Austria (Table 4, Fig. 4). In
general, the variation in $k$ values were lower in Austria, while the variation in $S$ were lower in
Sweden. It is possible that the ash correction, that was made for the Swedish but not the Austrian
sites, may partly explain this difference. Indeed, the average mass loss after 90 days at the Swedish
sites for Green and Rooibos tea was higher with about 60 and 30%, respectively, whereas it was
only about 45 and 15% for the Austrian sites (data not shown). When recovering litter dry matter
from the soil, soil-contamination are often not negligible. In our study, the ash-content determined
on the Green and Rooibos tea bags for the Swedish site represented 15±6 and 10±4 %, respectively
(data not shown).



*Influence of climate and soil properties*

In previous studies using the TBI approach, it was shown that climate played a significant role on decomposition in a temperate biome (Djukic et al., 2018), but when comparing several different biomes, climatic conditions were of relatively low importance (Fanin et al., 2020). In Boreal soils, Althuizen et al. (2018) found that increased temperatures enhanced $k$, whereas increased precipitation decreased $k$ across years. Despite that many studies have showed a positive correlation between precipitation and decomposition rates (Pimentel et al., 2019; García Palacios et al., 2016), precipitation did not have a huge impact in our study according the random forest analysis. On the other hand, $Re_{wat}$ showed great importance (Fig. 6). It is because this variable includes nonlinearities due to its shape according to which decomposition increases with soil moisture and then decreases at high soil water content due to oxygen limitation of microorganisms (Moyano et al., 2013).

In general, higher $k$ values were observed when the aridity index (AI) was lower. AI was identified by the random forest regression model being an important variable affecting the rate of decomposition (Fig. 6). Soils from more arid and warmer sites are associated with lower SOC (Kerr and Ochsner, 2020; Ontl and Schulte, 2012). With increasing aridity, the biological processes that drive C and N inputs and fluxes in ecosystems may be impaired, which may result in decreasing soil C and N stocks (Jiao et al., 2016; Reynolds et al., 2007).

The random forest models showed that the decomposition rate $k$ was mostly affected by climate, in particular when considering the TBI periods combined (Fig. 6a). The lower predictive power of the models when considering the 60 and the 90 days TBI periods separately can explain the higher number of variables considered, due to less defined effects to be identified by the model. This is suggested also by the decrease in the overall node purity of the models using only 60, or 90 days





data to explain *k*. When using the combined dataset, the model was instead explaining a relatively
large part of the variance ($R^2$=0.735) and with a much higher node purity, while employing very
few and only climatic-related parameters.
For practical reasons the teabags in our study were buried in the soil during the growing season,
corresponding to a period when the soil biological activity is highest (Bolinder et al., 2013). When
burying the teabags during the growing season, the difference in climate between sites are
attenuated, in particular with respect to air temperature. For example, the MAT at the Swedish
most northerly (SE5 and SE9) and southerly (SE2) sites are 4.1 and 9.0 $^{\circ}$C, respectively, whereas
corresponding mean air temperatures ($MAT_{TBI}$) during our study were 14.3 and 16.2 $^{\circ}$C,
respectively.
The TBI *S* parameter was also dependent on climate. Indeed, the random forest model identified
climatic parameters as the main factors affecting *S* in both Austria and Sweden during all evaluated
periods. In particular, $Re_{clim}$ and $Re_{temp}$ often showed significant negative correlations, which
implies a negative impact of air temperature on *S*. However, raw climatic variables, such as
precipitation and temperature were only weakly correlated with *S*. This is probably due to
nonlinear processes, which are considered in the ICBM climate-dependent soil biological activity
calculations such as $Re_{wat}$ (as discussed above). Furthermore, since litter decomposition dynamics
is influenced by multiple factors that interact and change over time (Bradford et al., 2016), the
relationships are not always linear. Random forest models that we fitted to the data are more
efficient capturing such combinations and interactions of factors, and can detect relationships that
would not be detectable by linear approaches.
The stabilization factor *S* expresses the degree by which the labile fraction of the plant material is
decomposed. Therefore, it is not surprising that more variables come into play to define it. In





particular, we noticed the influence of edaphic factors, of which pH was the most important, but
also SOC concentration, C:N ratio and clay were also identified as good predictors. In addition,
agronomic factors were also influencing *S*, where soil management treatment and crop types were
the most important. A study conducted by Fu et al. (2021) suggested that pH, nutrient availability
and soil compaction were the main reasons contributing to the differences in litter decomposition.
The net effect of pH is not clear since it modifies both SOC decay kinetics and productivity
simultaneously (Paradelo et al., 2015). Nevertheless, the impact of pH on SOC kinetics seems
clear in our study with a maximum effect at around neutral pH (Liao et al., 2016).

**Conclusion**
Our results show that both TBI *k* and *S* parameters were sensitive to management practices in
agroecosystems in Austria and Sweden. We were observing significant differences for some of the
treatments in all categories of LTEs. Notably, for the effect of crop residue incorporation, organic
amendments and N fertilization, crop types and tillage systems. In the Austrian LTEs, application
of green manure + municipal compost showed a higher *S* compared to the application of other
organic amendments. Incorporation of crop residues and high N fertilization also increased *k*. In
the Swedish LTEs, it was shown that combined management practices with both farmyard manure
and mineral NPK resulted in higher *k* and *S* compared to no manure and no NPK applications,
whereas growing spring cereals instead of leys increased *k* but did not change *S*. For both countries,
tillage systems with deep reduced tillage practices presented higher *k*, and shallow reduced tillage
presented higher *S*. However, these effects were also site or year dependent within a given country.
Climatic conditions had the most important impact on the decomposition rate *k* and the
stabilization factor *S*, but also SOC, C:N ratio and clay content were good predictors of the TBI



parameters. Generally, the correlations with raw climatic variables such as precipitation and
temperature were quite poor. Better relationships were found when nonlinearities due to
interactions between climatic and edaphic conditions were accounted for. Our results bring
knowledge and answers under how a wide range of soil management practices affect soil
decomposition jointly to soil and climatic conditions. We recommend the TBI approach for further
LTE studies evaluating soil decomposition dynamics.

**Data availability**
Data can be provided by the authors upon request.

**Author contribution**
MRG, MB and TS wrote and prepared the manuscript draft; MB, TK and TS supervised and led
the research; MB, OA, HS, JKF, AS, AS and TS developed the methodology; MRG and LM
analyzed the data. All authors reviewed and edited the manuscript.

**Competing interests**
The contact author has declared that none of the authors has any competing interests.

**Acknowledgements**
Financial support was provided by the Swedish Farmers' Foundation for Agricultural Research,
grant number O-18-23-141. Part of this research has been done in the framework of the EJP SOIL
that has received funding from the European Union's Horizon 2020 research and innovation
programme: Grant agreement No 862695.



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



**Tables**

Table 1. Experimental sites in Austria. Site acronym, location name, year of TBI measurements,
category and type of LTE including its duration, main crop cultivated during the TBI
measurements and management treatments

| Site | Location | Year | Category* | Experiment† | Age | Crop | Treatments§ |
|------|----------|------|-----------|-------------|-----|------|-------------|
| AT1 | MUBIL | 2014 | CB | OA | 11 | Luzerne | GM, FW, FYM, BS |
| AT2 | MUBIL | 2015 | CB | OA | 12 | Wheat | GM, FW, FYM, BS |
| AT3 | Ritzlhof | 2015 | CB | IF-N & OA+N | 24 | Maize | 0, 40, 90, 120 N CFW, CGM, CS, CSS+80N |
| AT4 | Rottenhaus | 2016 | CB | CR with NPK | 31 | Maize | CRR, CRI |
| AT5 | Rutzendorf | 2015 | CB | CR with NPK | 34 | Wheat | CRR, CRI |
| AT6 | Rutzendorf | 2016 | CB | CR with NPK | 35 | Maize | CRR, CRI |
| AT7 | Breitstetten | 2015 | SF | IF-N with PK | 40 | Wheat | 60, 90, 120, 145 N |
| AT8 | Breitstetten | 2016 | SF | IF-N with PK | 41 | Wheat | 60, 90, 120, 145 N |
| AT9 | Haringsee | 2015 | SF | IF-N with PK | 40 | Wheat | 60, 90, 120, 145 N |
| AT10 | Fuchsenbigl | 2016 | SF | IF-K with NP | 61 | Maize | 0, 150, 300 K |
| AT11 | Rottenhaus | 2016 | SF | IF-K with NP | 63 | Maize | 0, 150, 300 K |
| AT12 | Haringsee | 2016 | SF | IF-N without PK | 41 | Maize | 0, 60, 120, 180 N |
| AT13 | Zinsenhof | 2016 | SF | IF-N without PK | 41 | Maize | 0, 60, 120, 180 N |
| AT14 | Zissersdorf | 2016 | SF | IF-N without PK | 41 | Maize | 0, 60, 120, 180 N |
| AT15 | Fuchsenbigl | 2015 | TS | TS | 27 | Maize | CT, SRT, DRT |
| AT16 | Fuchsenbigl | 2016 | TS | TS | 28 | Wheat | CT, SRT, DRT |

* CB: carbon balance practices; SF: soil fertility; TS: tillage systems.
† OA: organic amendments; IF-N & OA+N: N inorganic fertilization and organic amendments plus mineral N; CR &
IF-P with NK: crop residues and P inorganic fertilization with 90 and 40 kg ha$^{-1}$ of N and K, respectively; IF-N with
PK: N inorganic fertilization with 55 and 180 kg ha$^{-1}$ of P and K, respectively; IF-N without PK: N inorganic
fertilization; IF-K with NP: K inorganic fertilization with 120 kg N ha$^{-1}$; TS: tillage system.
§ GM: green manure; FW: municipal compost and green manure; FYM: farmyard manure; BS: biogas slurry; CFW:
compost food waste with 175 kg N ha$^{-1}$; CGM: compost green manure with 175 kg N ha$^{-1}$; CS: compost slurry with
175 kg N ha$^{-1}$; CSS: compost sewage sludge with 175 kg N ha$^{-1}$; CRR: crop residues removed; CRI: crop residues
incorporated; CT: conventional tillage; SRT: shallow reduced tillage; DRT: deep reduced tillage.




Table 2. Experimental sites in Sweden. Site acronym, location name, year of TBI measurements
conducted in 2016, type of LTE including its duration, main crop cultivated during the TBI
measurements and management treatments

| Site | Name | Experiment* | Age | Crop | Treatments† |
|------|------|-------------|-----|------|-------------|
| SE1 | Börgeby | CMP | 59 | Sugar beet | FYM/NPK |
|  |  |  |  | Sugar beet | 0 FYM/0 NPK |
| SE2 | Ekebo | CMP | 59 | Sugar beet | FYM/NPK |
|  |  |  |  | Sugar beet | 0 FYM/0 NPK |
| SE3 | Högåsa | CMP | 50 | Ley production year | FYM/NPK |
|  |  |  |  | Ley production year | FYM/0 NPK |
|  |  |  |  | Spring oilseed | 0 FYM/NPK |
|  |  |  |  | Spring oilseed | 0 FYM/0 NPK |
| SE4 | Kungsängen | CMP | 53 | Oat | FYM/NPK |
|  |  |  |  | Oat | 0 FYM/0 NPK |
| SE5 | Röbacksdalen | CMP | 47 | Barley | FYM/NPK |
|  |  |  |  | Barley | FYM/0 NPK |
| SE6 | Vreta Kloster | CMP | 50 | Ley production year | FYM/NPK |
|  |  |  |  | Ley production year | FYM/0 NPK |
|  |  |  |  | Spring oilseed | 0 FYM/NPK |
|  |  |  |  | Spring oilseed | 0 FYM/0 NPK |
| SE7 | Lanna | ROT | 35 | Oat | SC |
|  |  |  |  | Ley production year | L |
| SE8 | Lönnstorp | ROT | 36 | Barley | SC |
|  |  |  |  | Ley establishment year | L |
| SE9 | Röbacksdalen | ROT | 36 | Barley | SC |
|  |  |  |  | Ley establishment year | L |
| SE10 | Säby | ROT | 46 | Wheat | SC |
|  |  |  |  | Ley establishment year | L |
| SE11 | Lanna | TS | 34 | Winter wheat | CT, DS |
| SE12 | Säby | TS | 11 | Barley | CT, SRT, DRT, DS |
| SE13 | Ultuna | TS | 19 | Barley | CT, DRT |

* CMP: combined management practices; ROT: rotation systems; TS: tillage systems.
† FYM/NPK: maximum amount of farmyard manure and maximum doses of NPK; 0 FYM/0 NPK: no manure and no
NPK application; FYM/0 NPK: maximum amount of farmyard manure and no NPK application; 0 FYM/NPK: no
manure and maximum doses of NPK; SC: spring cereal; L: ley; CT: conventional tillage; DS: direct seeding; SRT:
shallow reduced tillage; DRT: deep reduced tillage.




Table 3. Annual climatic characteristics for the Austrian and Swedish sites during the entire year of measurements and only during the TBI period (days) corresponding to the period between the date of placement and the last retrieval date of the tea bags.

| Site | TBI period | TAP | $TP_{TBI}$ | MAT | $MAT_{TBI}$ | PET | $PET_{TBI}$ | AI | $AI_{TBI}$ |
|---|---|---|---|---|---|---|---|---|---|
| | days | mm | | °C | | mm | | | |
| *Austria* | | | | | | | | | |
| AT1 | 99 | 756 | 294 | 12.5 | 20.3 | 811.4 | 414.0 | 1.07 | 1.4 |
| AT2 | 61 | 516 | 65 | 12.6 | 18.8 | 605.1 | 141.3 | 1.17 | 2.2 |
| AT3 | 80 | 622 | 161 | 12.2 | 20.0 | 682.9 | 295.1 | 1.09 | 1.8 |
| AT4 & 11 | 84 | 939 | 384 | 11.2 | 20.5 | 763.4 | 306.1 | 0.81 | 0.8 |
| AT5 | 63 | 516 | 82 | 12.6 | 19.4 | 605.1 | 148.6 | 1.17 | 1.8 |
| AT6 | 96 | 735 | 226 | 12.0 | 21.6 | 854.2 | 394.0 | 1.16 | 1.7 |
| AT7 | 59 | 516 | 81 | 12.6 | 19.3 | 605.1 | 137.4 | 1.17 | 1.7 |
| AT8 | 84 | 735 | 240 | 12.0 | 17.1 | 854.2 | 334.2 | 1.16 | 1.4 |
| AT9 | 60 | 516 | 81 | 12.6 | 19.2 | 605.1 | 139.6 | 1.17 | 1.7 |
| AT10 & 16 | 92 | 735 | 205 | 12.0 | 21.4 | 851.2 | 378.9 | 1.15 | 1.8 |
| AT12 | 87 | 735 | 203 | 12.0 | 21.8 | 854.2 | 367.4 | 1.16 | 1.8 |
| AT13 | 84 | 939 | 327 | 11.2 | 20.5 | 763.4 | 304.3 | 0.81 | 0.9 |
| AT14 | 82 | 735 | 154 | 12.0 | 21.3 | 854.2 | 319.6 | 1.16 | 2.1 |
| AT15 | 81 | 516 | 103 | 12.6 | 22.5 | 605.1 | 212.9 | 1.17 | 2.1 |
| *Sweden* | | | | | | | | | |
| SE1 | 92 | 436 | 141 | 9.2 | 16.7 | 444.6 | 235.1 | 1.02 | 1.7 |
| SE2 | 91 | 681 | 270 | 9.0 | 16.5 | 578.5 | 297.8 | 0.85 | 1.1 |
| SE3 | 91 | 458 | 142 | 7.9 | 13.3 | 618.0 | 306.4 | 1.35 | 2.1 |
| SE4 | 93 | 429 | 126 | 7.1 | 15.3 | 640.0 | 367.0 | 1.48 | 2.9 |
| SE5 & 9 | 92 | 526 | 164 | 4.1 | 14.3 | 396.2 | 218.2 | 0.75 | 1.3 |
| SE6 | 91 | 433 | 158 | 7.9 | 13.3 | 617.5 | 301.8 | 1.16 | 1.9 |
| SE7 & 11 | 91 | 412 | 115 | 7.4 | 12.7 | 514.2 | 276.4 | 1.3 | 2.4 |
| SE8 | 90 | 610 | 224 | 9.4 | 14.4 | 500.2 | 265.7 | 0.82 | 1.2 |
| SE10 & 12 | 91 | 429 | 126 | 7.1 | 15.3 | 358.5 | 155.7 | 0.84 | 1.2 |
| SE13 | 92 | 429 | 126 | 7.1 | 15.3 | 639.1 | 377.8 | 1.5 | 3.0 |

TAP: total annual precipitation; $TP_{TBI}$: total precipitation during TBI period; MAT: mean annual temperature; $MAT_{TBI}$: mean temperature during TBI period; PET: total annual potential evapotranspiration; $PET_{TBI}$: potential evapotranspiration during TBI period; AI: annual aridity index (PET divided by TAP); $AI_{TBI}$: aridity index during TBI period.





Table 4 – Mean values of decomposition rate ($k$) and stabilization factor ($S$) for the TBI approach
after 60 and 90 days of incubation period.

| Incubation | Mean TBI parameters | |
| --- | --- | --- |
| | $k$ | $S$ |
| *Sweden* | | |
| 60 days | 0.0160 | 0.296 |
| 90 days | 0.0152 | 0.267 |
| *Austria* | | |
| 60 days | 0.0152 | 0.429 |
| 90 days | 0.0115 | 0.423 |







**Figures**

**FIGURE 1**

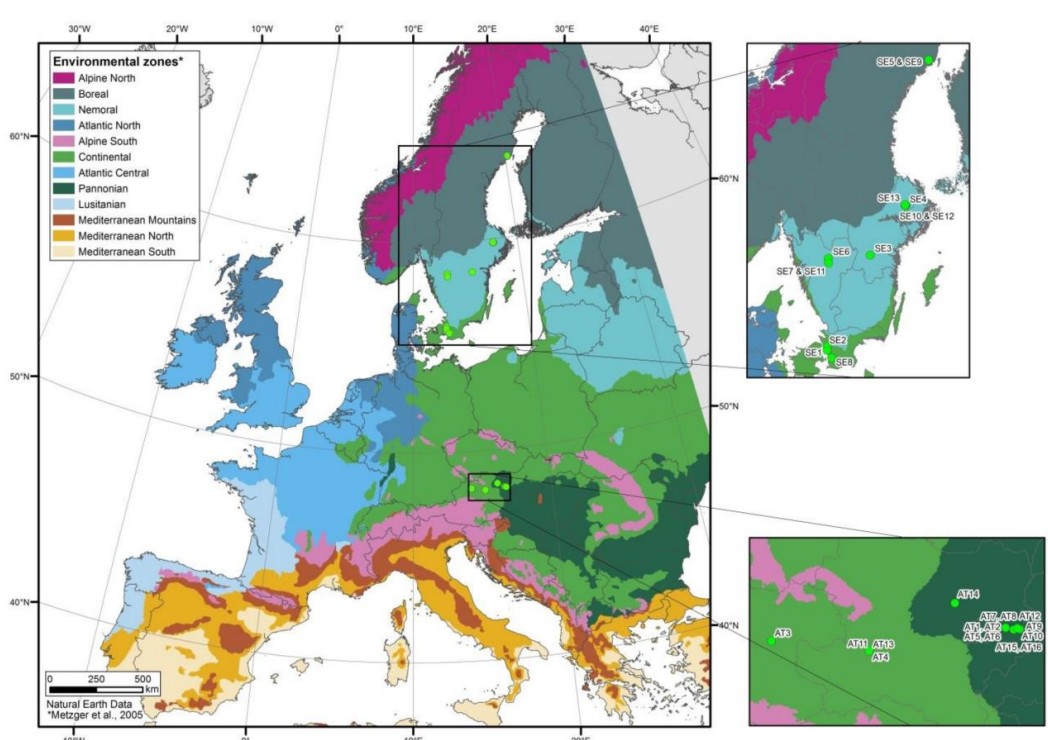




**FIGURE 2**

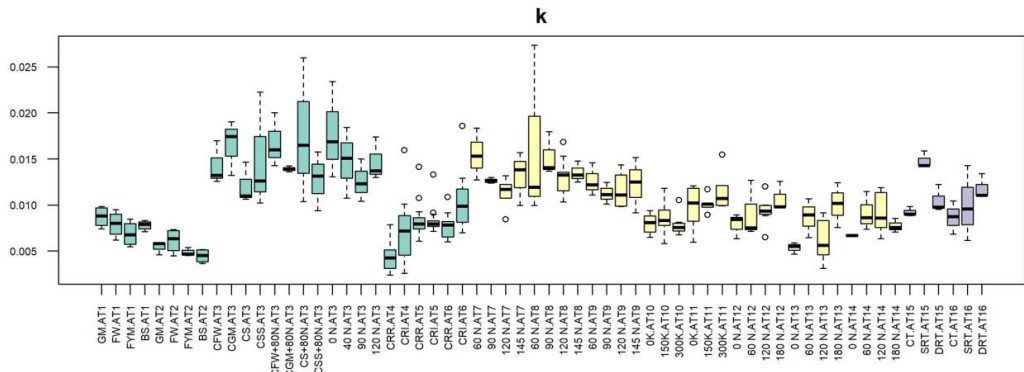

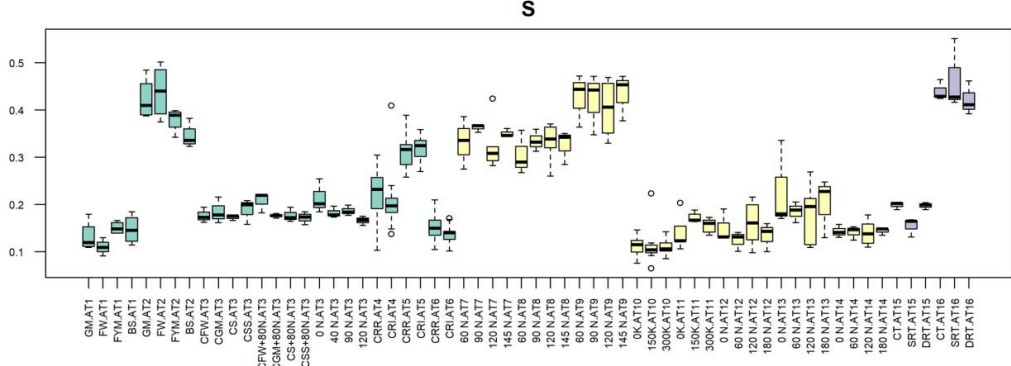




**FIGURE 3**

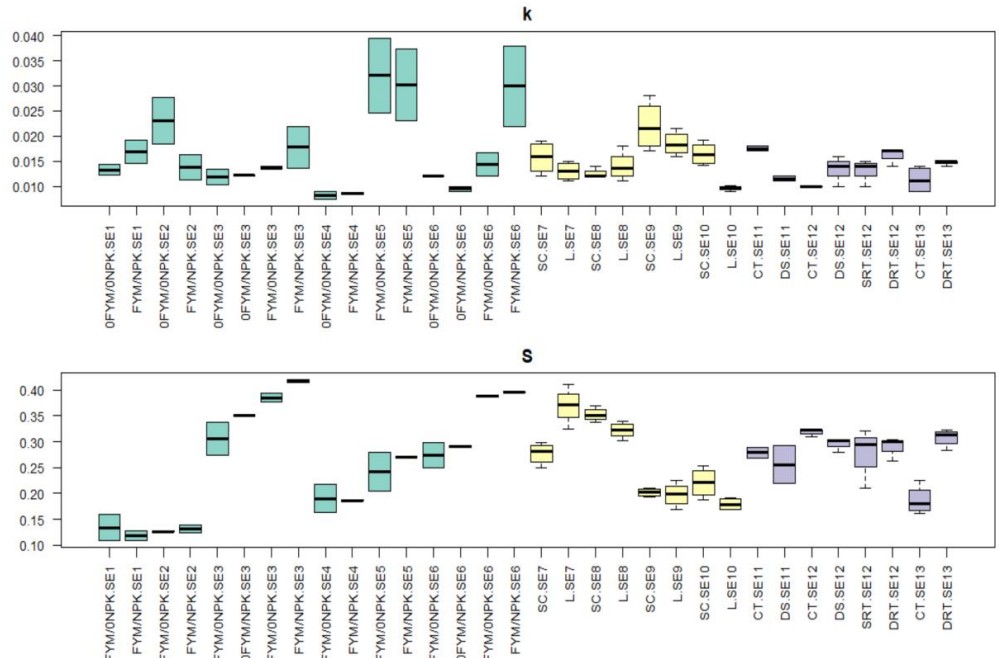






**FIGURE 4**

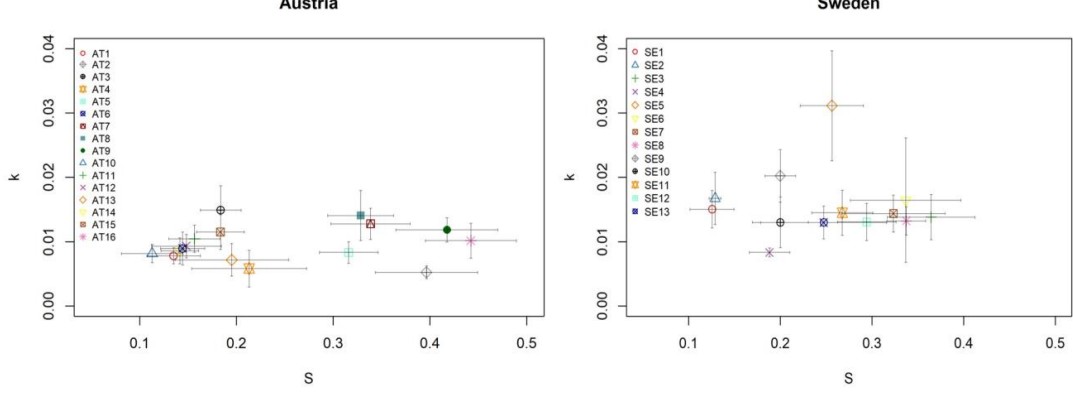




**FIGURE 5**

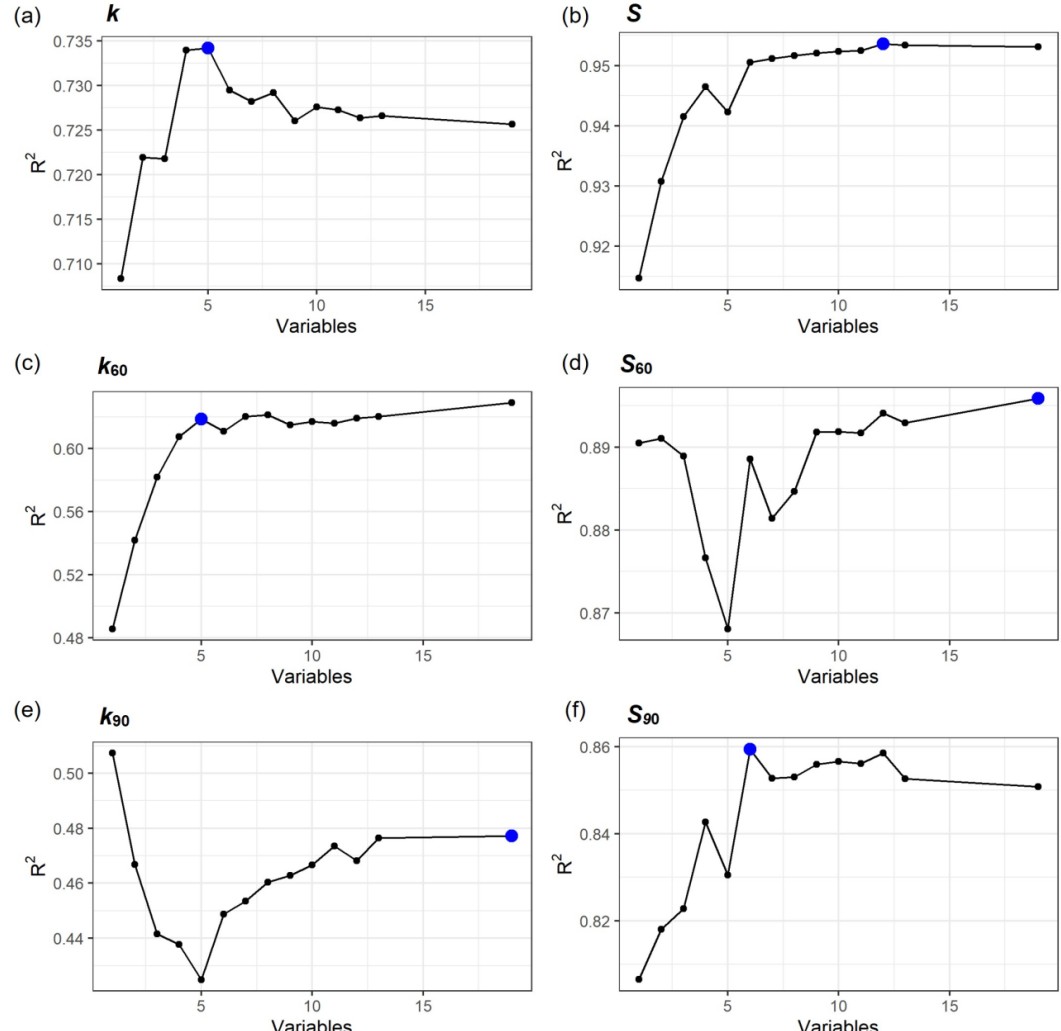






**FIGURE 6**

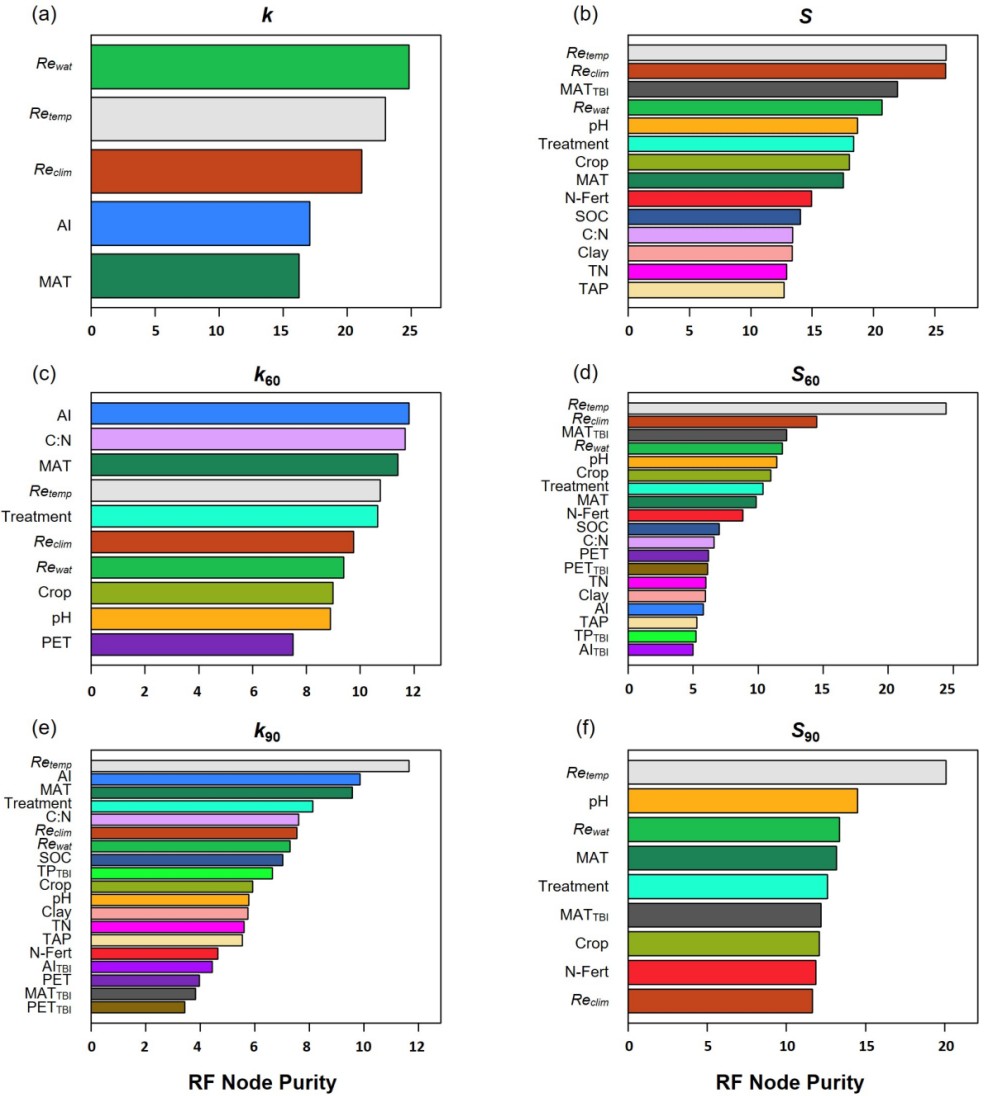




**Figure caption**

Figure 1 - Location and environmental zone of the Austrian and Swedish sites.

Figure 2 - Average decomposition rate ($k$) and stabilization ($S$) after the 90 days TBI period for
each treatment and site in Austria. The extents of the box indicate 25th and 75th percentiles, and the
lines represent the 50th percentile. Whiskers represent the 10th and 90th percentiles and outliers are
given as open symbols. Green boxes: Carbon balance (CB) experiment; yellow boxes: Soil fertility
(SF) experiment; purple boxes: tillage systems (TS) experiment. Site AT1 shows results from
2014. Sites AT2, AT3, AT5, AT7, AT9, and AT15 show results from 2015. And sites AT4, AT6,
AT8, AT10, AT11, AT12, AT13, AT14, and AT16 show results from 2016.

Figure 3 - Average decomposition rate ($k$) and stabilization (S) after the 90 days TBI period for
each site and treatment in Sweden. The extents of the box indicate 25th and 75th percentiles, and
the lines represent the 50th percentile. Whiskers represent the 10th and 90th percentiles.

Figure 4 - Distribution of the mean decomposition rate constant ($k$) and the stabilization factor ($S$)
after the 90 days TBI period for each site in Austria and Sweden. Errors bars represent standard
deviation.

Figure 5 - Variables selection procedure to identify the optimal number of variables to explain the
variance of $k$ and $S$ considering the combined dataset (60 and 90 days TBI period) with a Random
Forest model. The blue point represents the optimal model. a) and b) Variables affecting $k$ and $S$





over all sampling times; c) and d) Variables affecting $k$ and $S$ after 60 days; e) and f) Variables
affecting $k$ and $S$ after 90 days.

Figure 6 - Relative importance of the variables used by each optimized Random Forest model to
predict the variance in the $k$ and $S$ parameters for the combined dataset (60 and 90 days TBI
period) in Austria and Sweden. The higher the Node purity, the higher the importance of such
variable. a) and b) Variables affecting $k$ and $S$ in all times; c) and d) Variables affecting $k$ and $S$
after 60 days; e) and f) Variables affecting $k$ and $S$ after 90 days.