# Peer review of "Evaluating the Tea Bag Index approach for different management practices in"

_EGUsphere, 2023_

## Editor Comment (EC1)

**Evaluating the Tea Bag Index approach for different management practices in**

**agroecosystems using long-term field experiments in Austria and Sweden**

[revised manuscript text omitted]

---

## Author Response (AR1)

**Answers to the Editor and the Reviewers**

Dear Dr. Ingrid Lubbers,

I am pleased to submit the revised version of our manuscript entitled **"Evaluating the Tea Bag Index approach for different management practices in agroecosystems using long-term field experiments in Austria and Sweden",** for consideration to be published in **SOIL.**

We now modified all the main topics you pointed out and some of the most important modifications are highlighted below:

1. Our focus in this manuscript was to evaluate the decomposition after ~90 days, using the standard TBI method. For this reason, we decided to remove all the results including time-series and/or combined datasets and show only the final incubation value, since they were creating certain confusion. Furthermore, the 90 days incubation is the official protocol of the TBI approach. We have thoroughly explained the rationale behind these considerations in the materials and methods sections. We also changed some figures accordingly, and we removed Figure S1.

2. We performed the PCA analysis which was indeed quite helpful to understand some results. This is now showed in the new Figure 6, and discussed throughout the text.

3. We have changed the results and the discussion topic in order to express the main findings in a better way, and we modified the entire text reducing the acronyms making the text clear and easier to follow.

4. We have pointed out the criticism and issues about the chosen method.

We thank the Editorial Board and the Reviewers for their interest in our manuscript and detailed feedback. We are looking forward to receive your comments.
Additional details on how these comments have been addressed appears in the response to the reviewers.

**Reviewer 1:**

I want to congratulate the authors on an impressive study that took a lot of samples and generated a very rich dataset. I think that the questions posed in the introduction are interesting and of scientific relevance. Due to the large dataset and the many interesting conclusions that could be drawn from that, I do think it should eventually be published in SOIL. However, I think that the study needs to be improved in a number of aspects, before it can be accepted for publication. Also, I think the study focuses too much things that are well known (effect of temperature and moisture) and only superficially touches the more interesting aspects (effect of other soil properties such as SOC, CN – and that of management). The authors could probably make more of this highly interesting

dataset, if the methods were refined. I specified a few suggestions on how that could be done, below.

R: Dear Dr. Moritz Laub, thank you very much for your interest in our manuscript and for all your good suggestions. We have now modified and improved the whole text.

Comments to specific parts:

Introduction: Overall well written – not much to improve there.

Just one thing: The term fertilization should be replaced by fertilizer application (throughout the text), because fertilization refers to a specific mechanism in sexual reproduction and is just not the right word.

R: Ok, modified.

Methods:

The methodology lacks some important details, especially on the rate modifiers from ICBM, on the way statistics were applied and on dataset split for different forms of analyses. Also, with the high number of experiments, it is especially important to structure their description and the results by some common denominator across experiments. The way it is now, I find it very hard to follow.

R: Concerning details about the calculation of rate modifiers we now included in the text references to the calculation. That calculation is rather complex, with lot of steps, and documenting it fully in this manuscript would not be feasible without making it too long. However, we substantially changed the paragraph describing the rate modifiers from ICBM and it should be clear for the reader how they were used. Furthermore, the calculation steps are now fully documented as the manual of an R package stored here: https://github.com/ilmenichetti/reclim. The package has been also submitted to CRAN but it will take some time before it is available there. It can nevertheless already be installed and the documentation is fully accessible, both in form of vignette and on-line R help. Please see the main changes below, which is now in Lines 270-293:

"We calculated a climate-dependent soil biological activity parameter ($Re_{clim}$), by using mean daily air temperature, total precipitation, and potential evapotranspiration (PET) data in pedotransfer, soil water balance and biological activity functions. Compared to raw climatic data alone, this parameter is integrating the effect of climate, soil and crop properties. It is calculating as the product of a soil temperature ($Re_{temp}$) and relative water content ($Re_{wat}$) factor with a daily time step (i.e., $Re_{clim} = Re_{temp}$ x $Re_{wat}$), which is thereafter averaged to give an estimate of soil biological activity for a given time period. These two factors are derived from soil temperature and soil moisture response functions expressing the activity of decomposers and their relative effect on the decay rates of organic materials in the arable layer of agricultural soils. Briefly, $Re_{temp}$ is calculated from air temperature and leaf area index using an empirical model (Kätterer and Andrén, 2009), while $Re_{wat}$ is calculated using pedotransfer functions for simulating the soil water balance and a function for estimating PET. In addition to air temperature and leaf area index, calculations of $Re_{wat}$ also involve the use of daily climatic data for precipitation, wind

speed, air humidity, and solar radiation, as well crop types and yields, soil texture and SOC content. For details see Bolinder et al., 2008, and Fortin et al., 2011) and information in the following R package used for the calculations: https://github.com/ilmenichetti/reclim (Please refer to the package documentation). The Reclim concept can be used for quantifying regional differences in soil biological activity alone (Bolinder et al., 2013; Andrén et al. 2007) or integrated as a parameter in the ICBM SOC model. In the latter case, it is adjusting the decomposition rates of both C inputs to soil from crop residues (e.g., straw) and that of the more stable SOC (Andrén and Kätterer, 1997; Andrén et al., 2004). In this study, we used the concept of $Re_{clim}$ and the $Re_{temp}$ and $Re_{wat}$ factors to test if the product of soil-temperature and relative water content better explained the variation in k and S than the two latter alone, and to determine if they also better explained this variation compared to using only raw climatic data."

I am further wondering, if you could not make much more of the data if you included a temperature function (and maybe moisture function) in the TBI calculation directly (e.g. W(t)= a e-k t f(T) + (1-a), where f(T) is a temperature function as used in most biogeochemical models. That may help you to distill out better the effect of management. It is well known that that temperature and moisture is important, so I would think that the question what is important for a temperature and moisture adjusted k and S of the TBI would be of much higher scientific relevance. After all models such as ICBM are around for many decades and have such rate modifiers since long.

R: We did not consider the possibility of using the re_clim parameter and the soil temperature and soil moisture factors as a kinetic scaling because we would have lost compatibility with all the rest of the literature around the TBI index.

The main purpose of the TBI approach lies in the possibility of comparing TBI results between studies and with a vast TBI library around the globe. Therefore, we decided to adhere to the TBI original standard approach, and letting the correlation analyses, the Random Forest machine learning model (and now also an PCA) determine the relative effects of the different climatic variables considered in this study.

The retemp and rewat are integrating the effect soil and crop types, that is why we also include temperature and precipitation as separate variables in our analysis (i.e., to see if retemp and rewat performs better). We modified the text and clearly identified the rationale behind this. Please see the changes below, which is now in Lines 270-293:

"We calculated a climate-dependent soil biological activity parameter ($Re_{clim}$), by using mean daily air temperature, total precipitation, and potential evapotranspiration (PET) data in pedotransfer, soil water balance and biological activity functions. Compared to raw climatic data alone, this parameter is integrating the effect of climate, soil and crop properties. It is calculating as the product of a soil temperature ($Re_{temp}$) and relative water content ($Re_{wat}$) factor with a daily time step (i.e., $Re_{clim} = Re_{temp}$ x $Re_{wat}$), which is thereafter averaged to give an estimate of soil biological activity for a given time period. These two factors are derived from soil temperature and soil moisture response functions

expressing the activity of decomposers and their relative effect on the decay rates of organic materials in the arable layer of agricultural soils. Briefly, $Re_{temp}$ is calculated from air temperature and leaf area index using an empirical model (Kätterer and Andrén, 2009), while $Re_{wat}$ is calculated using pedotransfer functions for simulating the soil water balance and a function for estimating PET. In addition to air temperature and leaf area index, calculations of $Re_{wat}$ also involve the use of daily climatic data for precipitation, wind speed, air humidity, and solar radiation, as well crop types and yields, soil texture and SOC content. For details see Bolinder et al., 2008, and Fortin et al., 2011) and information in the following R package used for the calculations: https://github.com/ilmenichetti/reclim (Please refer to the package documentation). The Reclim concept can be used for quantifying regional differences in soil biological activity alone (Bolinder et al., 2013; Andrén et al. 2007) or integrated as a parameter in the ICBM SOC model. In the latter case, it is adjusting the decomposition rates of both C inputs to soil from crop residues (e.g., straw) and that of the more stable SOC (Andrén and Kätterer, 1997; Andrén et al., 2004). In this study, we used the concept of $Re_{clim}$ and the $Re_{temp}$ and $Re_{wat}$ factors to test if the product of soil-temperature and relative water content better explained the variation in k and S than the two latter alone, and to determine if they also better explained this variation compared to using only raw climatic data."

I found it a bit hard to follow all the different experiments and how they are structured. The tables help to understand them but the text is a bit unstructured. It seems a bit random what details are described about each of them vs what details omitted. I would suggest you either make the description less details and rely fully on the tables or you describe every experiment in the exact same order (e.g., 1) cropping system, 2) tillage system 3) inputs 4) exports etc.).´

R: It is challenging to present results from so many different long-term experiments. This implies that there are unavoidably some variation in what type of information that is available or not from a given experiment/site. We believe that we now have a clear structure of how the experiments are presented (i.e., by different categories), and we have presented the available information that were most common to all experiments.

I think that statistical tests using ANOVA are not correctly accounting for the fact that there are a lot of samples in the dataset that are not independent (e.g. from different years at the same sites). I would think that a mixed linear model would be much more suitable, given such nested structure of data.

R: We have treated the data from the different years at the same sites as independent observations. In the sense that it is not a strict time series of measurements. There were mostly different crops between years for the same site, and there were different tea bags buried each year. So, we have not really repeated measurements over time as such.

We also improved and better explained the statistical analyses. See now in Lines 261-269, the paragraph now reads: "Analysis of variance (ANOVA) for each experiment category

(i.e., CB, SF, TS, CMP and ROT) was performed to analyze the effects of the treatments and the differences between sites on $k$ and $S$ separately for both countries. When the treatments were identical within the same experiment category, sites were used as a random effect with a mixed ANOVA to test the average treatment effect, mean values were used as replicates to test the differences between sites. The Tukey's test ($p < 0.05$) was used for comparing the same treatments and the same sites using R software version 4.2.2. We have treated the data from different years at the same sites in Austria as independent observations, in the sense they are not a time series of measurement. Interactions between site and treatment were considered."

I am a bit concerned that samples from Sweden and Austria were treated differently (measuring ash content versus not). This means that the results are not comparable – at minimum this difference in treatment should be a covariate in random forest. Also, for Austria data – how can you assure that differences, especially those between sites, but due to soil structure maybe even those between treatments are not just more or less soil particle sticking to the tea?

R: We understand this concern but we do not believe this is a problem. For the Austrian sites, we followed the standardized TBI protocol and removed adhering soil particles carefully. For the Swedish experiments, we measured the ash content because three of the sites had a high clay content and the removal of adhering soil particles can be more problematic. However, the ash content was low. Furthermore, this aspect does not influence the relative comparisons of k and S between treatments. This information has now been added to the text in the materials and methods section, which can be seen in lines 215-221 and 232-244, and now reads: "For Austria, four bags of each tea were used and placed side by side at a distance of 2 to 3 cm. Each tea bag was properly identified and buried in the soil at 8 cm depth. The TBI incubation period from placement to last retrieval averaged 80±13 days (Table 3) due to logistic issues for collection. After collecting, the tea bags were oven-dried at 70°C for 48 hours after removal of adhered soil particles according to the standardized protocol by Keuskamp et al. (2013). After drying, the tea bags were opened and the tea content was weighted. The ash content was not determined." and "After measuring the remaining dry matter, the decomposition rate ($k$) and stabilization factor ($S$) for both countries were calculated according to the TBI presented by Keuskamp et al. (2013). This standardized method that is using single measurements after an incubation period in the soil of 90-days have received some criticism. For instance, Mori (2022) and Mori et al. (2022) showed that this incubation period is not always long enough for the mass loss of green tea to reach a plateau, and further suggested that time-series mass loss data of rooibos tea is also required to respect the underlying assumptions of the TBI method. Time-series (15, 30, 60 and 90-days) of green and rooibos tea were available for all the Swedish sites but only at one Austrian site (16, 26, 62 and 91-days at AT16). The incubation period was consistently always 90-days for the Swedish sites, and only shorter than that (i.e., about 60-days incubation period) for a few of the Austrian sites (Table 3). To have as uniform comparisons as possible

between the two datasets, we only used the last measurement for both countries for calculating $k$ and $S$. The purpose of using the time-series for testing the underlying TBI assumptions was beyond the scope of this paper."

To me, it is not really clear how Retemp and Rewat were calculated and used in this study. This is poorly described. Did you actually run ICBM or did you just use the typical Retemp and Rewat functions and calculate them yourself? As specified above, I think this is actually a good idea, you just need to make it more clear what it is that you actually did.

R: Please see our previous explanations above. We actually did not run ICBM, we were only using the climate parameter and soil temperature and moisture factors. We rewrote this paragraph accordingly and this should now be clear (please see lines above).

Results:
The results are also presented in a somewhat difficult to follow fashion. With so many abbreviations, it would help to write out at least the treatments instead of describing K, FW, CRR… And then you talk about AT1, AT2 ect. and suddenly about the Fuchsenbiggl LTE – this is also confusing. Also, is it not clear if all the differences you describe are significant – and Fig 2 and 3 also do not show this.

R: We changed the writing throughout and reduced the use of abbreviations in the text, and we believe that it is clearly mentioned when the differences were significant.

Discussion:
In the discussion, I am not sure if the comparison of differences in k between different sites is all that meaningful. As long as k and S are not moisture/temperature corrected those have probably the highest influence and the rest is mostly speculation. Also I would start with the major findings of the RF model (what are the most important factors determining k and S) before going into the details of each individual experiment.

R: It is true that moisture and temperature likely have a large influence, but this "limitation" is somewhat inherent to the method. In order to have a standard approach to gather data from researchers (it is also devoted to applications in citizen science) to assess broader relevance globally, the calculations need to be simple, and there is a trade-off between simplicity and complexity. The purpose of our study was to present and discuss the results using the standard TBI protocol.

Line specific comments:
L44 Were the random forest regressions done by country or with the joint dataset?
R: It was done jointly. We now improved it in the text.

L58 I would rather call it emission capacity, because as far as I know, most agricultural soils are actually loosing SOC.

R: Actually, in well managed agricultural systems, soils have the potential to mitigate greenhouse gas emissions by storing it, as shown in the referenced articles.

L62-64 Maybe refine this statement a bit. Microbes are actually a main contributor to stable SOC (https://www.nature.com/articles/s41561-022-01100-3), wheres mere C input is not stable at all.
R: Ok, modified.

L70 What is exactly is meant by "extended process"? And why is it only decomposition not stabilization? (The focus on decomposition is very teabag focused).
R: We understand that the term "decomposition" also encompasses the SOC stabilization processes. But we modified the text so that is now clear to the reader.

L85 I think it would be suitable to display the differential equation used to derive S and k and to shortly describe the fitting process
R: We do not believe that it is necessary since we do not change anything in those equations and especially because it would make the manuscript too long. The equations are fully described in the original article (Keuskamp et al., (2013).

L95 I would argue that, because TBI is not normalized to mean annual temperature, it is more influenced by temperature than soil quality.
R: That is a good point of view.

L96 The argument that agroecosystems have the least been studied does not match the fact that you have much more references there???
R: That is correct, it was confusing. We modified the text to better explain this point of view (please see lines 97-105).

L102 I think it would be good to elaborate a bit more what LTEs were used. E.g., similar soil conditions, different management.
R: We have improved some details in the text and in the table S1.

L107 distinguish instead of detect? Ok
L120/121 Please clarify: What are C balance practices? Residue removal vs incorporation? And how can an LTE be used to study soil fertility if treatments have the same soil?
R: We clarified this issue in the text in Lines 126-135 and 136-141, which now reads: "Six experiment categories involved carbon balance practices (CB) focusing on organic matter inputs such as compost and crop residues, eight sites were studying soil fertility (SF) in terms of differences in mineral N and P fertilization, and two sites examined tillage systems (TS). The sites are located in several agricultural areas across the country (Fig. 1), with diverse soil textures (Table S1) and variable crop types (Table 1) and climatic

characteristics (Table 3) during the years of TBI measurements. More details for some of the sites are available in specific publications: AT3 (Spiegel et al., 2018; Lehtinen et al., 2017; Tatzber et al., 2015; Aichberger and Söllinger, 2009), AT4 to AT6 (Spiegel et al., 2018; Lehtinen et al., 2014), AT15 and AT16 (Tatzber et al., 2015; Spiegel et al., 2007). In addition, Sandén et al. (2018) puts the Austrian LTEs in the context of other European LTEs." and "The purpose of the Austrian C balance LTEs was threefold: i) to assess the sustainability of stockless vs. livestock-keeping organic farming management on soil and crop traits (AT1, AT2), ii) to investigate the effects of compost amendments on soil and crops (AT3), and iii) to compare crop residue incorporation and removal (AT4-AT6). Compost amendment LTE and crop residue incorporation LTEs also included mineral fertilizer application, whereas AT1 and AT2 only focused on different organic fertilizer treatments."

L163 Please clarify if the dairy production treatments also include cash crops
R: Ok, modified in the text.
L171 You have not described the rotation in detail, so the specifics for some crops are a bit confusing here.
R: Modified in the text.

L205 What do you mean by identified? Uniquely labeled? And was there only one bag buried in each treatment at each site?
R: We identified each tea to differ Green tea from Roiboos tea and the replicates, since after the experimental period the label could have been destroyed. We used 4 tea bags per experimental unit (replicate), as described in the text.

L212 Did you assure in any other way that no soil stuck to the tea litter if ash content was not determined?
R: Yes, we followed the original standardized TBI methodology and adhering soil particles were removed. This is now specified in the text in the Lines 215-221, as: "For Austria, four bags of each tea were used and placed side by side at a distance of 2 to 3 cm. Each tea bag was properly identified and buried in the soil at 8 cm depth. The TBI incubation period from placement to last retrieval averaged 80±13 days (Table 3) due to logistic issues for collection. After collecting, the tea bags were oven-dried at 70°C for 48 hours after removal of adhered soil particles according to the standardized protocol by Keuskamp et al. (2013). After drying, the tea bags were opened and the tea content was weighted. The ash content was not determined."

L214 The four bags for AT sites were not described above. Or did I miss it?
R: Actually, it was described on line 206, which now is line 215.

L218 So for SE sites you did determine soil contamination? I find it a bit problematic for comparability that this was done for SE but not AT.

R: This is correct and the rationale for doing that in Sweden is now explained in the manuscript. Since the ash content is unknown for Austria, we can never make the results perfectly comparable in that regard. However, we do not consider this a problem (please see our previous replies on the same aspect).

L227ff It is not clear which measurement was done at which level of detail. I would suspect TOC at the plot level, texture at experiment level, maybe – but please specify.
R: Texture was done by site. Otherwise the soil data is done for each field plot studied.

L238 You have repeated measurements in your data. How did you assure that by this you did not artificially inflate statistical power? Have you averaged repeated measurements? Also did you average the 4 replicates per treatments? I think you need to do this, or you use a mixed linear model, probably best with a nested structure such as experiment/block/treatment
R: We made an average of the four replicates and when the difference was significant we applied Tukey's test for comparing treatments or sites (see Tables S2 and S3). We have treated the data from different years for the same sites as independent observation. This is now better explained in the text in Lines 261-269, and can be read as: "Analysis of variance (ANOVA) for each experiment category (i.e., CB, SF, TS, CMP and ROT) was performed to analyze the effects of the treatments and the differences between sites on $k$ and $S$ separately for both countries. When the treatments were identical within the same experiment category, sites were used as a random effect with a mixed ANOVA to test the average treatment effect, mean values were used as replicates to test the differences between sites. The Tukey's test ($p < 0.05$) was used for comparing the same treatments and the same sites using R software version 4.2.2. We have treated the data from different years at the same sites in Austria as independent observations, in the sense they are not a time series of measurement. Interactions between site and treatment were considered."

L242 do you mean you adjusted your k by it? I am confused because the whole article is about litter and now you talk about SOC pools.
R: This paragraph was now rewritten and there should be no confusion about this.

L243 It is probably just a rate modifier, i.e., f(T) * f(m), so actually just a function of temperature and moisture. Arguably those two are influenced by climate, crop and soil properties. Maybe best just give the function of the rate modifiers
R: This paragraph was now rewritten and there should be no confusion about this. Please, see comments above.

L247 Should Retemp not be a function of soil temperature?
R: Yes, Retemp is a function of soil temperature, this is now clearly described in the materials and methods section.

L253 Which variables? What do you mean by "more accurate results"?
R: We added to the text, the variables are: MAT, MTTBI, TAP, TPTBI, PET, PETTBI, AI, AITBI, TxP, Reclim, Retemp, Rewat, pH, Clay, SOC and C:N ratio. Random forest is a more accurate type of correlation than the simple ones. We explained this better in the following lines.

L256ff I think you can shorten the RF description to the most essential and just cite appropriate literature
R: Ok, modified in the text.

L266 I think you should cite some paper describing how "node purity" and "Gini index" are defined
R: It was now better explained in the Material and Methods.

L275/6 What is the difference between TN and N? MATTBI should be MTTBI no?
R: Indeed TN and N are not the same. It may seem confusing but we need to simplify the equation. The acronyms are explained in the lines following the equation.

L303 So is AT5 and AT6 actually the same experiment just in different years?
R: Yes, and also with different crops. We now highlighted this in the text.

L339 You have not described this decomposition model in the methods.
R: We removed the time-series and decomposition model data from this paper, since this is not the main objective and could create confusion.

L355 Fewer than what?
R: The text was modified.

L359-61 Yes, and this is exactly why I would advocate that you also try the RF on a temperature and moisture corrected k and S value from TBI, e.g. by using Retemp and Rewat. Because the finding that temperature and moisture are important is really nothing new, but the influence of management factors and soil types much more interesting.
R: We added some more discussion on these aspects.

L430 – how much ash was actually there in Sweden? Does is give you an indication that your AT data may be strongly biased? How much is the ash content of the litter before burial?
R: The ash content in Sweden represented about 10% of the dry matter for Rooibos tea after 90 days, while it was about 15% for Green tea, as shown in the text. We do not believe Austrian data were biased because of that, please see our previous responses on

that aspect above. The ash content of green and rooibos tea prior to burial represents 2 and 4%, respectively (see Keuskamp et al. 2013).

L438 I would start the discussion with this section
R: We agree with your suggestion and we now changed the order of the Discussion topic.

L445 I would argue that precipitation and Rewat are likely strongly correlated. So if you have both in your dataset, you cannot conclude that precipitation is not important if not selected by RF.
R: We wrote this in a better way.

L458/9 I do not understand what you want to express here.
R: Ok, it was improved in the text.

L462 Where in your results do you display node purity? Have I missed it?
R: In figure 5, where the x axis is shown in this unit.

L485 To me, this is the most interesting part of your results. You should put more emphasis on this and less on the well-known temperature/moisture effects. It would also be interesting in which direction SOC, CN etc influences S?
R: We improved it in the text and also added a PCA analysis, as suggested by the Reviewer 2, then we can realize the directions and influences in k and S. Please see Fig. 6 (below) and the new text in the Discussion section.

Figure 6 - Principal component analysis showing how the sites in Austria and Sweden differ based on the variables. PC1 and PC2 are the first two components, explaining most variance. The loadings (black arrows) are the weight of each variable in defining each principal component. The size of the arrows can tell how much they contribute defining this space, while the direction is their contribution on each axis.

[Figure]

tap: total annual precipitation; TxP: temperature x precipitation factor; pet: potential evapotranspiration; mat: mean annual temperature; tn: total soil nitrogen; soc: total soil organic carbon; cn: soil C:N ratio; ai: aridity index.

Table 4: I think you should state mean k and S per country and tea type and also give measures of uncertainty (e.g. SD).
R: We cannot separate it for tea types since we solely focus on the TBI parameters. We added the SD values as suggested. Please see below:

Table 4 – Mean values of decomposition rate ($k$) and stabilization factor ($S$) for the TBI approach after the incubation period.

|  | Mean TBI parameters | |
| --- | --- | --- |
|  | $k$ | $S$ |
| *Sweden* | $0.0160 \pm 0.01$ | $0.247 \pm 0.14$ |
| *Austria* | $0.0115 \pm 0.004$ | $0.228 \pm 0.11$ |

Fig 2: The font size is very small and hard to read.
R: We tried to increase the font size but it was not possible due to the figure size. But we increased the figure a little bit and we are going to suggest to the editors put Figures 2 and 3 in a Landscape layout.

Fig 2 and 3 should further indicate which treatments differ significantly between treatments at the same site (this is only in the text, so far).
R: We understand it would be easier to follow, but those figures are already huge and full of information, they are one way to see what is happening. The significant differences between treatments are properly shown in tables S2 and S3.

Citation: https://doi.org/10.5194/egusphere-2023-1229-RC1

**Reviewer 2:**
The study "Evaluating the Tea Bag Index approach for different management practices in agroecosystems using long-term field experiments in Austria and Sweden» investigates the effects of various types of management on decomposition processes estimated by the recently widely used TBI approach. This is an attempt to generalize a lot of field experiment data from two regions and with different kinds of experiments (organic and mineral fertilization, cropping and tillage systems) of different duration and under different crops. The structure of initial data complicates the analysis and further presentation of the results, and sometimes the text is difficult to follow, especially when the authors refer to site abbreviations AT1, AT2, etc.
R: Dear Dr. Tatiana Elumeeva, thank you for your interest and your good suggestions. Indeed, it is challenging to present results from so many experiments but we believe that now we have a clear structure of how they are presented (i.e., by different categories). We reduced the use of other abbreviations throughout the text, which should now facilitate the reading of site abbreviations.

I think the use of multivariate methods would be useful for visualization of the results, may be the PCA ordination of sites (treatments) using soil and climate variables to reveal the main gradients of environment, and highlighted types of treatment and directions of STBI and kTBI changes.
R: It was a very good suggestion. We made the PCA analysis which was very helpful to understand some results. Please see Figure 6 (below). Also see the Material and Methods and throughout the text.

Figure 6 - Principal component analysis showing how the sites in Austria and Sweden differ based on the variables. PC1 and PC2 are the first two components, explaining most variance. The loadings (black arrows) are the weight of each variable in defining each principal component. The size of the arrows can tell how much they contribute defining this space, while the direction is their contribution on each axis.

[Figure]

tap: total annual precipitation; TxP: temperature x precipitation factor; pet: potential evapotranspiration; mat: mean annual temperature; tn: total soil nitrogen; soc: total soil organic carbon; cn: soil C:N ratio; ai: aridity index.

Below are the minor comments on the text:

Line 95: "Camellia sinensin" – Camellia sinensis

R: Ok., modified

Line 97: "Aspalanthus linearis" replace by Aspalathus linearis

R: Ok, modified.

Lines 238–240: It would be better to describe the ANOVA a bit more. Here it seems the ANOVA was applied to the total data set, but when reading Table S2 it seems that Tukey's test was applied mostly to the single sites and to the 2 or 3 sites with the same treatments in Austria and groups of experiments in Sweden.

R: As in Austria we had different treatment categories within the experiment, we treated the data separately. And when it had the same treatments, we treated them jointly as shown in table S2, e.g., AT4 and AT6, AT5 and AT6, AT7 and AT9, AT10 and AT11, and AT12, AT13 and AT14. But in Sweden, the different sites for the same Experiment had the same treatments, so we treated the data jointly as well. We improved the text for better understanding, and can now be read in Lines 261-269 as: "Analysis of variance (ANOVA) for each experiment category (i.e., CB, SF, TS, CMP and ROT) was performed to analyze the effects of the treatments and the differences between sites on $k$ and $S$ separately for both countries. When the treatments were identical within the same experiment category, sites were used as a random effect with a mixed ANOVA to test the average treatment effect, mean values were used as replicates to test the differences between sites. The Tukey's test ($p < 0.05$) was used for comparing the same treatments and the same sites using R software version 4.2.2. We have treated the data from

different years at the same sites in Austria as independent observations, in the sense they are not a time series of measurement. Interactions between site and treatment were considered."

Line 303: "Comparing years for the same experiment type, AT5 (2015) had higher S than AT6 (2016)." – is that due to years or due to spatial heterogeneity between sites?
R: We do not believe that there is a large heterogeneity between sites since it was the same site, at different years. It is possible due to climatic conditions and maybe due to the crop type. We improved the discussion in Lines 485-493, which can now be read as: "In the tillage system experiment at Fuchsenbigl in Austria in 2015 (AT15, with maize) and 2016 (AT16; with wheat), the shallow reduced tillage showed significantly higher $k$ than deep reduced tillage and conventional tillage, only in 2015, indicating that shallow soil tillage stimulated decomposition that particular year. This was likely due to climatic conditions, since 2015 was slightly drier and warmer, furthermore maize straw has lower C:N ratio than wheat straw and tends to decompose faster. Some studies showed faster decomposition under conventional tillage than under reduced tillage practices (e.g., Lupwayi et al., 2004). However, Kainiemi et al. (2015) found a decrease in soil respiration in conventional tillage compared to shallow tillage in temperate regions, which directly implies a lower decomposition (and lower $k$)."

Lines 482-483: "The stabilization factor S expresses the degree by which the labile fraction of the plant material is decomposed." – Doesn't it expresses the degree by which the labile fraction was transferred to recalcitrant one?
R: Indeed, from teatime4science webpage (FAQ): Stabilization (S) is the degree to which litter breaks down, and k is the rate by which this occurs, or to be more precise, the degree and rate by which the labile fraction of the plant material is decomposed.

Line 802: "IF-K with NP: K inorganic fertilization with 120 kg N ha-1;" – What about P?
R: We missed it. We now added this information.

Figures 2 and 3 contain too many values, so they are difficult to understand and unclear, which values should be compared.
R: We understand. Unfortunately, it is a huge figure illustrating the interaction between site and treatment for k and S. We separated the experiments categories by color. We improved this as much as we could by increasing the size of this figure. The significant differences between treatments and between sites are shown in Tables S2 and S3.

The following citations are absent in the list of references:
Line 97: Pino et al. 2021
Line 125: Spiegel et al., 2018; Lehtinen et al., 2017
Line 126: Lehtinen et al., 2014

Line 127: Spiegel et al., 2007
The following references are not cited in the text:
Line 537: Al-Kaisi et al., 2008.
Line 593 Cornwell et al, 2008.
Line 600: Couteaux et al, 1995.
Line 608: Didion et al., 2016.
Line 614: Domínguez et al., 2014.
Line 620: Dubeux Jr et al., 2006.
Line 623: Duddigan et al. 2020.
Line 626: Elumeeva et al., 2018.
Line 636: Food and Agriculture Organization (FAO), 2005.
Line 643: Freschet et al., 2012.
Line 654: IPCC 2021
Line 654: IPCC 2022
Line 687: Kätterer et al., 2014.
Line 697: Kohmann et al., 2019.
Line 714: Martin et al., 2020.
Line 733: Pingel et al., 2019.
R: Sorry for this mistake. We corrected all of these.

Citation: https://doi.org/10.5194/egusphere-2023-1229-RC2

**Reviewer 3:**
The authors present an interesting, valuable, large dataset on decomposition of rooibos and green tea in different long-term agricultural field experiments across Austria and Sweden. It is an important topic to find indicators suitable to assess the influence of different agricultural management practices on soil quality. The manuscript is well written and, in most points, easy to follow. Considering some aspects in more detail, could help clarify main conclusions.
Especially in the Swedish dataset, there seems to be more information available regarding this topic, because decomposition was measured after 15, 30, 60 and 90 days. Thus, k values can be modelled individually per site via a decay function as was done in Figure S1 for all sites together. But I strongly recommend doing this analysis separately per site. See papers suggested already by Mori and also our study Middelanis et al. 2023.
To combine this with suggestions of reviewer 1 would be ideal to focus more on management effects instead of climate effects. Instead of showing general differences between Austria and Sweden in Figure 4 it would be more interesting to focus on management aspects such as fertilizer application, tillage system ect.
In general, I find it hard to understand why you mixed up data obtained after 60 days of decomposition and those obtained after 90 days. The labile fraction of rooibos tea is not

yet decomposed (even after 90 days), as you show in Figure S1. To my understanding this does not make sense. But you spend much space in your manuscript to present these data. What is your conclusion based on the combined data analysis versus the individual data analysis?

R: Dear Dr. Ute Hamer, thank you very much for your interest in our paper and for your good suggestions. We understand this criticism about the combined dataset and 60 days. Our main focus in this manuscript was to evaluate the decomposition after ~90 days, using the standard TBI method. Furthermore, we do not have data for the time-series (15, 30, 60 and 90 days) for all sites. For this reason, we decided to remove all the results including time-series and/or combined datasets and show only the final decomposition value. We have now thoroughly explained the rationale behind these considerations in the materials and methods sections in Lines 232-244, and now can be read as: "After measuring the remaining dry matter, the decomposition rate ($k$) and stabilization factor ($S$) for both countries were calculated according to the TBI presented by Keuskamp et al. (2013). This standardized method that is using single measurements after an incubation period in the soil of 90-days have received some criticism. For instance, Mori (2022) and Mori et al. (2022) showed that this incubation period is not always long enough for the mass loss of green tea to reach a plateau, and further suggested that time-series mass loss data of rooibos tea is also required to respect the underlying assumptions of the TBI method. Time-series (15, 30, 60 and 90-days) of green and rooibos tea were available for all the Swedish sites but only at one Austrian site (16, 26, 62 and 91-days at AT16). The incubation period was consistently always 90-days for the Swedish sites, and only shorter than that (i.e., about 60-days incubation period) for a few of the Austrian sites (Table 3). To have as uniform comparisons as possible between the two datasets, we only used the last measurement for both countries for calculating $k$ and $S$. The purpose of using the time-series for testing the underlying TBI assumptions was beyond the scope of this paper. ".

I miss a bit the discussion on the suitability of the random forest analysis and the conclusions which can be drawn from the results. Does it really make sense to include 17 variables as in Figure 5 b, 5 d and 5e? As far as I see in the Figure $R^2$ increases only marginally. What is the lesson which can be learned from the information that the factor treatment was selected as e.g. the 7th variable out of 17 explaining S after 60 days (Figure 6d)?

R: We believe it does make sense exactly to show that a lot of the variance can be explained with few variables (and we classify their importance later in fig 5). Those plots are offering to the reader information about the sensitivity to additional information of the models to predict each variable. The fact that k peaks very fast gives some hints that the information ending up being represented by that specific parameter is not correlated with many other covariates other than water content and temperature, while S is instead slightly more complex. Fig 5 offers information about how much each variable was contributing to predict each parameter, so how these are correlated, and RF

can utilize also categorical variables, this was one of the main reasons why we choose this technique after briefly considering multivariate dimensional reductions.

When I looked up the factor treatment in the Material and Method section it notes that there are 30 levels of treatment (line 281), which ones? For your first analysis you distinguished between CMP, ROT and TS. Why do you not consider this in your random forest models?

R: The treatment levels are all the different treatments involved in the studies from both countries. This information is provided in the column "Treatment" of Tables 1 and 2. We do not consider the Experiment category because the focus on this study was to compare the treatments within each category.

Line 281: Why is maize not included in the crop factor of your random forest model?

R: You are right, maize was included and we corrected the text accordingly.

I find it quite interesting that "In Sweden, litter decomposition differed more between treatments than between sites" (line 41, 42), although the gradient in soil properties seems to be larger in Sweden than in Austria. According to Table S1 clay content in Sweden ranges between below 5 to 50 %. Would be quite interesting to figure out under which circumstances treatment effects become more visible. Do you have an explanation for this?

R: We agree. This suggestion was based on Figures 2 and 3 but they were not well conclusive so we decided to remove them.

Table S1: Please add information on soil type, as indicated in the heading! To which soil depth do the data refer to? And to which treatment? Sites have different treatments and addition of FYM might for example increase SOC… Please, add more precise information for all treatments of specific sites. Please also check soil texture and clay content, e.g. a silty clay with 5.6 % clay at SE 4 seems to be strange.

R: Unfortunately we do not have information for all the soil types so we removed it from the heading.
The soil depth to these data is 20 cm and this information was now added to the text.
As you suggested, we now have separated the soil characteristics for each treatment in the table. We also checked all the values again and clay content in SE4 is not 5.6, but 56%, due to a typing mistake. Thank you. Please see Table S1.

Line 107 to 109: check sentence structure.

R: Ok, modified.

Line 204f: how many tea bags have been burrowed per site? Only 1 green and 1 rooibos tea bag? Please indicate.

R: We had four replicates of each tea. This was indicated in the following text.

Line 234: was the pH in Swedish soil samples also measured in 0.01 M CaCl2?
R: It was pH in water and this was now specified in the text.

Line 303f: how can you be sure that this is a year effect and not an effect of the crop species?
R: That is a relevant suggestion. We improved it in the discussion section in Lines 485-493, and can be now read as: "In the tillage system experiment at Fuchsenbigl in Austria in 2015 (AT15, with maize) and 2016 (AT16; with wheat), the shallow reduced tillage showed significantly higher $k$ than deep reduced tillage and conventional tillage, only in 2015, indicating that shallow soil tillage stimulated decomposition that particular year. This was likely due to climatic conditions, since 2015 was slightly drier and warmer, furthermore maize straw has lower C:N ratio than wheat straw and tends to decompose faster. Some studies showed faster decomposition under conventional tillage than under reduced tillage practices (e.g., Lupwayi et al., 2004). However, Kainiemi et al. (2015) found a decrease in soil respiration in conventional tillage compared to shallow tillage in temperate regions, which directly implies a lower decomposition (and lower $k$).".

Line 402ff: again: how can you be sure that differences are due to climatic conditions and not to different effects of maize versus wheat (there is a lot of literature around showing beneficial effects of wheat compared to maize)
R: That is corrected. We now improved the discussion. Please see the comment above.

Line 456ff: this is rather a repetition of results than a real discussion.
R: Ok, it was modified.

Table 1: It is a bit confusing using different site numbers for the same site in different years with a different crop in rotation! I suggest to number according to locations and year, eg. AT1_1 and AT1_2 for location AT 1 and year 1 and year 2, respectively.
R: We did not change this; we contend that the current numbering of sites is the most appropriate to use for an easy reading. This is also supported by the fact that different years for the same sites are treated separately in the statistical analysis. Furthermore, we believe that increasing the acronymous sizes could generate more confusion.

Figure 3: why are Whiskers missing in some cases, is n too small? Please indicate the number of replicates per treatment!
R: The Whiskers are not missing; they are just too small at some points. We used 4 replicates per treatment which is now well explained in the text.

Middelanis, T., Pohl, C. M., Looschelders, D., & Hamer, U. (2023). New directions for the Tea Bag Index: Alternative teabags and concepts can advance citizen science. Ecological Research, 1–10. https://doi.org/10.1111/1440-1703.12409

Citation: https://doi.org/10.5194/egusphere-2023-1229-RC3

**Community comment (1):**

As a premise for evaluating this paper, it is important to note that, in the context of the TBI approach, the parameter k is defined as the decomposition constant characterizing the asymptote model describing the decomposition curve of rooibos tea. Meanwhile S is computed as the ratio of the stabilized fraction to the hydrolysable fraction in green tea. TBI approach offers the advantage of determining both k and S using a single set of mass loss values acquired during an incubation period of approximately 90 days. Importantly, this approach assumes two essential assumptions, and only when both of these assumptions are met, the need for time-series data is obviated. First, the most portion of the hydrolysable fraction in green tea is decomposed within the initial 90 days (the first assumption). Secondly, the stabilization factor S, denoting the ratio of the stabilized to the total hydrolysable fractions, remains the same for rooibos tea as it does for green tea (the second assumption)

Given the aforementioned premises, my first comment on the authors' work pertains to their utilization of time-series data within their research. Despite the acquisition of these time-series data in their work, the authors omitted a rigorous assessment of their alignment with the fundamental assumptions underpinning the TBI approach. Instead, the authors simply computed the parameters k and S following the TBI method. I strongly recommend a thorough examination to ensure that these data comply with the specified assumptions.

R: Dear Dr. Taiki Mori, thank you for your interest in our manuscript and for your good suggestions. We understand this criticism. To test these assumptions was beyond the scope of this manuscript. Our focus in this manuscript was to evaluate the decomposition after 90 days, using the standard TBI method. We do not have the time-series data for all the sites. For this reason, we decided to remove all the results including time-series and show only the final decomposition value (after around 90 days). We have now thoroughly explained the rationale behind these considerations in the materials and methods sections in Lines 232-244, and can be read as: "After measuring the remaining dry matter, the decomposition rate ($k$) and stabilization factor ($S$) for both countries were calculated according to the TBI presented by Keuskamp et al. (2013). This standardized method that is using single measurements after an incubation period in the soil of 90-days have received some criticism. For instance, Mori (2022) and Mori et al. (2022) showed that this incubation period is not always long enough for the mass loss of green tea to reach a plateau, and further suggested that time-series mass loss data of rooibos tea is also required to respect the underlying assumptions of the TBI method. Time-series (15, 30, 60 and 90-days) of green and rooibos tea were available for all the Swedish sites but only at one Austrian site (16, 26, 62 and 91-days at AT16). The incubation period was consistently always 90-days for the Swedish sites, and only shorter than that (i.e., about 60-days incubation period) for a few of the Austrian sites (Table 3). To have as uniform

comparisons as possible between the two datasets, we only used the last measurement for both countries for calculating *k* and *S*. The purpose of using the time-series for testing the underlying TBI assumptions was beyond the scope of this paper.".

Moving on to my second point, it is imperative to note that these two premises have been challenged by previous works. My own research demonstrated that the application of an incubation study revealed a failure to meet these assumptions. This outcome holds particular significance (more robust than field study) due to the control of temperature conditions maintained throughout the incubation period (Mori 2022a). One could posit the argument that the TBI approach retains its utility for the comparative analysis of the "relative decomposition rate" across diverse ecosystem (or soil) types. However, another paper demonstrated the absence of a positive correlation between TBI-derived parameter k and the k values established through time-series data (Mori 2022a). Consequently, within the context of scientific research, it becomes apparent that the TBI approach may not be considered a suitable methodology.

Mori, T. (2022a). Validation of the Tea Bag Index as a standard approach for assessing organic matter decomposition: A laboratory incubation experiment. Ecol. Indic., 141, 109077.
Mori, T. (2022b). Is the Tea Bag Index (TBI) Useful for Comparing Decomposition Rates among Soils? Ecologies, 3, 521–529.

I acknowledge that a substantial number of publications employ the TBI protocol. Therefore, I refrain from endorsing the outright rejection of the publication of this paper. However, I do propose, as a minimum, two recommendations: (i) refraining from combining data derived from 60 days and 90 days when calculating TBI (mentioned in L271), and (ii) clearly mentioning the aforementioned limitation of the TBI method (i.e., the accuracy of the TBI method is challenged) in the paper to prevent any potential misinterpretation or misunderstanding by readers.

R: We agree and followed these two recommendations: i) we removed all the results combining 60 and 90 days (as described in our response above and we decided to show only the final burial date), ii) we clearly included this information in Lines 233-238, which reads as: "This standardized method that is using single measurements after an incubation period in the soil of 90-days have received some criticism. For instance, Mori (2022) and Mori et al. (2022) showed that this incubation period is not always long enough for the mass loss of green tea to reach a plateau, and further suggested that time-series mass loss data of rooibos tea is also required to respect the underlying assumptions of the TBI method.".

I would also propose refraining from constructing a graphical representation illustrating the relationship between parameters k and S (FIGURE 4). This recommendation is

grounded in the outcomes of a simulation study conducted by Mori et al. (2022), which demonstrated a kind of autocorrelation between k and S.

R: We agree that k and S can be a kind of autocorrelation. But this is a common approach used in TBI studies, and we think this figure is good to show in order to see what is happening.

Mori, T., Nakamura, R. & Aoyagi, R. (2022). Risk of misinterpreting the Tea Bag Index: Field observations and a random simulation. Ecol. Res., 37, 381–389.
Citation: https://doi.org/10.5194/egusphere-2023-1229-CC1

With kind regards,

Maria Regina Gmach
Institutionen för Ekologi - SLU
E-mail: gmachmr@gmail.com

---

## Editor Decision (ED1)

**Evaluating the Tea Bag Index approach for different management practices in**

**agroecosystems using long-term field experiments in Austria and Sweden**

Maria Regina Gmach[1]*, Martin A. Bolinder[1], Lorenzo Menichetti[1], Thomas Kätterer[1], Heide

Spiegel[2], Olle Åkesson[1,3], Jürgen Kurt Friedel[4], Andreas Surböck[4], Agnes Schweinzer[5], Taru

Sandén[2]

[1]Swedish University of Agricultural Science (SLU), Department of Ecology, Box 7044, 75007,

Uppsala, Sweden

[2]Austrian Agency for Health & Food Safety (AGES), Department for Soil Health and Plant

Nutrition, Spargelfeldstraße 191, A-1220 Vienna, Austria

[3]Lantmännen Lantbruk, Mariestadsvägen 104, 541 39 Skövde, Sweden

[4]University of Natural Resources and Life Sciences (BOKU), Department of Sustainable

Agricultural Systems, Institute of Organic Farming (IFÖL), Gregor-Mendel-Straße 33, A-1180

Vienna, Austria

[5]Easy-Cert services GmbH, Königsbrunner Straße 8, Austria

Corresponding author

* Maria Regina Gmach

Swedish University of Agricultural Science (SLU)

Uppsala, Sweden.

E-mail: gmachmr@gmail.com

Phone: +55 49991164271 / +49 17635956337

**Abstract**

Litter decomposition is an important factor affecting local and global C cycles. It is known that decomposition through soil microbial activity in ecosystems is mainly influenced by soil type and climatic conditions. However, for agroecosystems, there remains a need for a better understanding how management practices influence litter decomposition. This study examined the effect of different management practices on decomposition at 29 sites with long-term (mean duration of 38 years) field experiments (LTEs) using the Tea Bag Index (TBI) protocol with standard litter (Rooibos and Green tea) developed by Keuskamp et al. (2013). The objective was to determine if the TBI decomposition rate ($k$) and stabilization factor ($S$) are sensitive enough to detect differences in litter decomposition between management practices, and how they interact with edaphic factors, crop type and local climatic conditions. Tea bags were buried and collected after ~90 days in 16 Austrian and 13 Swedish sites. The treatments at Austrian LTEs focused on mineral and organic fertilizer application, tillage systems and crop residues management, whereas those in Sweden addressed cropping systems, mineral fertilizer application and tillage systems. The results showed that in Austria, incorporation of crop residues and high N fertilizer application increased $k$. Minimum tillage had significantly higher $k$ compared to reduced and conventional tillage. In Sweden, fertilized plots showed higher $S$ than non-fertilized plots and high N fertilizer had the highest $k$. Growing spring cereal lead to higher $k$ than forage crops. Random Forest regressions for Austria and Sweden jointly showed that $k$ and $S$ were mainly governed by climatic conditions, which explained more than 70% of their variation. However, under similar climatic conditions, management practices strongly influenced decomposition dynamics. It would be appropriate to apply the TBI approach in a more large-scale network on LTEs for agroecosystems, as an indicator to better assess, which of the management practices can best promote a higher soil C sink.

compared to what?

The next sentences are much clearer because comparisons are made.

so small k and large S? to store C the S needs to be high and the k low. or?

[revised manuscript text omitted]

---

## Author Response (AR2)

**Answers to the Editor and the Reviewers**

Dear Dr. Ingrid Lubbers,

I am pleased to submit the revised version of our manuscript entitled **"Evaluating the Tea Bag Index approach for different management practices in agroecosystems using long-term field experiments in Austria and Sweden",** for consideration to be published in **SOIL.**

We now modified all the minor topics you have pointed out and checked our conclusions, as you asked. We also have reviewed the English grammar and the text formatting.

We thank you very much, all the Editorial Board and the Reviewers for their feedback. We are looking forward to receive your new comments.

The answers to the Reviewers appear below:

**Reviewer #1:**

Overall, the article improved a lot and is easier to digest now. The effort to simplify the manuscript is greatly appreciated. I understand that challenge of summarizing such a large data set and I think you did a good job at simplifying it.

I can understand your argument why you did not want to include the moisture and temperature scalars into the TBI to make it comparable with other studies, even if I think it could make the results a bit more interesting. I see it was not the main focus of that manuscript, but just as food for though: You could probably even write another small article, just focusing on how management/soil properties affect a temperature/moisture normalized TBI.

One last thing: It seems to me now, that you removed the time series data completely from your analysis to simplify it and you only used the last data point. Would it then not make sense to also remove the description of the time series in M&M to avoid confusion? Or did you use the time series in the RF?

Dear Referee #1, thank you very much for your comments which were very valuable for the improvement of this manuscript. We will think about writing another small paper focusing in your suggestion, which seems quite interesting.

We did not remove all the descriptions about time series from the manuscript, since it explains the rationale behind how we decided to use (i.e., to combine) the datasets from Sweden and Austria and it is also part of our reply to reviewer #4. It should be clear that we have not used the time series in RF or any other analysis.

**Reviewer #2:**

The manuscript "Evaluating the Tea Bag Index approach for different management practices in agroecosystems using long-term field experiments in Austria and Sweden» has been substantially improved by the authors. I am completely satisfied by how they changed the draft and replied to my previous comments. May be, it would be useful to fit S and k values to the PCA axes and show them also at the Fig.6.

Dear Referee #2, thank you for your suggestion which help us to improve this manuscript. Regarding the PCA, we carefully considered your suggestion. In our analysis, the PCA was included only to examine similarities between the sites. We are considering the correlations between $S$ and $k$ with other variables using the other methods, which also allows us to better discriminate between the relative impact (which is proportional to correlation) on the two variables separately with different models. A RF model, although in a completely different way, does in practice something not too dissimilar that PCA finding the path of least variance (minimizing the residuals) in a multivariate variable space, as any multivariate regression technique would do including multiple linear regression. RF (as other machine learning algorithms such as ANN) is even more similar, since it is a nonlinear (actually even non-parametric) model. Although it is feasible including $S$ and $k$ in the PCA, we believe it would be redundant and, most importantly, would make the PCA less clear when it comes to understand the differences between the sites.

**Reviewer #4:**

Unfortunately, it appears that the authors have not adequately revised the manuscript. My previous comment may have been unclear.

The most important problem of the TBI stems from the unfounded assumption that S is uniform across both green and rooibos teas. This assumption (1) lacks theoretical grounding and (2) has been disproven by prior incubation studies. Consequently, this is not merely a minor discrepancy; rather, the calculated results are predicated upon erroneous assumptions, thereby leading to misleading outcomes.

Furthermore, the authors' justification for the acceptability of the correlation analysis between S and k, based solely on its prevalence in prior literature, is flawed. This rationale disregards theoretical objections, making it an unsound scientific approach to continue employing this methodology.

Dear Referee #4, your comments are very clear and we understand your criticism regarding the TBI method. We have specifically mentioned the issues that you raised in the materials and methods section, making these concerns available to the readers. However, as we also mentioned, it was beyond the scope of this manuscript to address these criticisms. We contend the TBI methodology is a well-known approach and suitable for the purpose of our study.

---

## Author Response (AR3)

**Answers to the Editor**

Dear Dr. Ingrid Lubbers,

I am pleased to submit the revised version of our manuscript entitled **"Evaluating the Tea Bag Index approach for different management practices in agroecosystems using long-term field experiments in Austria and Sweden",** for consideration to be published in **SOIL.**

We now modified and justified the minor topics you have pointed out.
Please see the answers below:

Editor's comments:
"Dear authors,

I think the manuscript is ready, but I have two small final question/comments. It concerns the conclusion in the abstract and the conclusions:
There is a sentence in the abstract and conclusions where it would make sense to make a comparison: "The results showed that in Austria, incorporation of crop residues and high N fertilizer application increased k." My comments are: compared to what? The next sentences are much clearer because comparisons are made."
R: We added the comparison and improved the sentence.

"The next point in the abstract and conclusions is the very last sentence: "... which of the management practices can best promote a higher soil C sink." My comments are: so small k and large S? to store C the S needs to be high and the k low. or? How does the TBI relate to the soil C sink?
Both points come back in the conclusions. Here I commented: Good to mention this here, to relate the TBI in the Austrian and Swedish LTEs to the timely issue of the potential (or not) to store C in arable soils. However, it is not clear how k and S may indicate a potential contribution to the C sink function of arable soils.
Would it be possible to give a direction to the C storage potential and the TBI outcomes of this study? Or why not?
I have attached my comments in the ms file."
R: We completely understand your perspective, and generally speaking, we can assume that a low $k$ and a high $S$ would imply a higher carbon storage in the soil. However, carbon storage is influenced by numerous factors, making it challenging to directly correlate TBI parameters with carbon sequestration. We felt hesitant about linking our findings to the carbon sink because we did not directly assess this aspect in our study. After discussing among the co-authors, we decided against going deeper into this topic, as it would involve speculation. As you suggested, we have revised both the abstract

and the conclusion to provide a more general statement. We believe our rationale is reasonable. Please, see the modifications below:

Lines 47-50: "It would be appropriate to apply the TBI approach in a more large-scale network on LTEs for agroecosystems, in order to improve its usefulness as an indicator for the effect of management practices on litter decomposition dynamics, particularly linking it with the potential for C storage."

Lines 561-564: "This also suggests that the TBI $k$ and $S$ parameters could serve as indicators of how different agricultural management practices influence the global carbon cycle via decomposition, a matter requiring further in-depth investigation."

We thank very much all the Editorial Board.
We are looking forward to receive your new comments.

Kind regards,
Maria Regina Gmach